# THE UNANTICIPATED ASYMMETRY BETWEEN PERCEPTUAL OPTIMIZATION AND ASSESSMENT

## ABSTRACT

Perceptual optimization is primarily driven by the fidelity objective, which enforces both semantic consistency and overall visual realism, while the adversarial objective provides complementary refinement by enhancing perceptual sharpness and fine-grained detail. Despite their central role, the correlation between their effectiveness as optimization objectives and their capability as image quality assessment (IQA) metrics remains underexplored. In this work, we conduct a systematic analysis and reveal an unanticipated **asymmetry** between perceptual optimization and assessment: fidelity metrics that excel in IQA are not necessarily effective for perceptual optimization, with this misalignment emerging more distinctly under adversarial training. In addition, while discriminators effectively suppress artifacts during optimization, their learned representations offer only limited benefits when reused as backbone initializations for IQA models. Beyond this asymmetry, our findings further demonstrate that discriminator design plays a decisive role in shaping optimization, with patch-level and convolutional architectures providing more faithful detail reconstruction than vanilla or Transformer-based alternatives. These insights advance the understanding of loss function design and its connection to IQA transferability, paving the way for more principled approaches to perceptual optimization. Code and models will be released publicly.

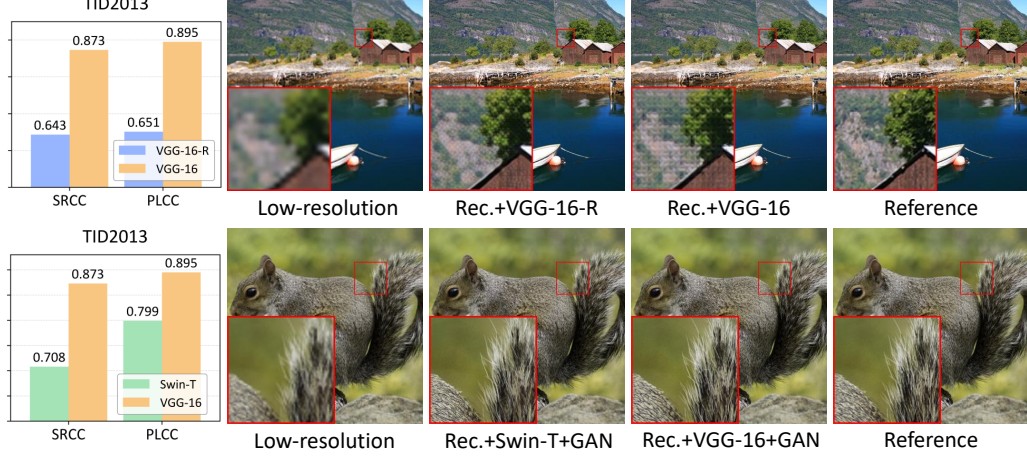

Figure 1: High-performing perceptual metrics on the IQA benchmark **FAIL** to consistently improve visual quality in perceptual optimization, whether adversarial loss is employed or not. We build perceptual metrics with diverse backbone architectures as optimization objectives, where Rec. denotes the $\ell_1$ norm, VGG-16-R indicates VGG-16 (Simonyan & Zisserman, 2014) with random weights, and Swin-T refers to the Swin Transformer (Liu et al., 2021).

## 1 INTRODUCTION

Amid the rapid progress of artificial intelligence, an increasing number of visual generation models have demonstrated the ability to synthesize high-resolution, photorealistic, semantically consistent, and visually detailed images (Lin et al., 2025; Batifol et al., 2025). A key factor driving these

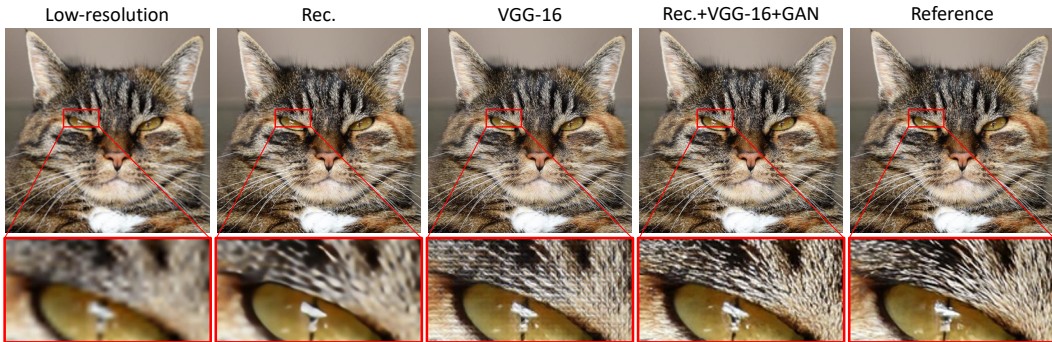

Figure 2: **Qualitative comparison** of SR results across different optimization objectives. While reconstruction and perceptual loss (VGG-16) tend to generate checkerboard artifacts, incorporating adversarial loss mitigates these distortions and yields more realistic textures.

advances is **perceptual optimization**, a fundamental technique that steers models toward generating visually realistic outputs while maintaining both structural fidelity and fine-grained textures relative to reference images (Wang et al., 2004). This approach has proven highly effective across a broad spectrum of vision tasks, spanning image restoration, enhancement, and generation (Wu et al., 2024a; Liang et al., 2023; Sauer et al., 2024), and continues to serve as a cornerstone for advancing both discriminative and generative visual modeling.

Fidelity metrics have historically been adopted as the primary optimization objectives, guiding models to align generated images with their references (Wang et al., 2004). Traditional perceptual optimization typically relies on pixel-wise reconstruction losses (*e.g.*, $\ell_1$ norm), which often produce over-smoothed results that diverge from human visual perception (HVS) (Liang et al., 2021) (see Fig.2). This shortcoming has motivated the adoption of deep full-reference image quality assessment (FR-IQA) metrics, such as LPIPS (Zhang et al., 2018), as auxiliary perceptual objectives to better preserve fidelity and produce visually consistent outputs (Ding et al., 2020). Nevertheless, as illustrated in Fig. 2, perceptual-oriented objectives can introduce irregular artifacts (*e.g.*, checkerboard patterns), often resulting from *surjective* feature mappings or the complexity of deep backbone architectures (Ding et al., 2021b). To address these visually unnatural artifacts, adversarial losses are incorporated as complementary objectives operating alongside fidelity metrics that enforce semantic consistency and global coherence, thereby forming the perceptual optimization paradigm that couples reconstruction- and perceptual-oriented fidelity terms with adversarial supervision (Wang et al., 2021; Esser et al., 2021; Sun et al., 2024; Tian et al., 2024).

Building on this integral formulation, recent studies have increasingly underscored the significance of fidelity metrics, particularly perceptual ones, thereby motivating the development of more advanced IQA methodologies that more effectively steer models toward high-fidelity and perceptually convincing image generation (Lao et al., 2022; Chen et al., 2024; 2025). This naturally raises a key question: *Do fidelity metrics with stronger IQA capability necessarily yield greater effectiveness when repurposed for perceptual optimization?* In parallel, given that discriminators are crucial for enforcing realism and suppressing artifacts during optimization, a complementary question arises: *Can their learned representations generalize sufficiently to serve as effective backbone initializations for IQA models?*

Motivated by these two questions, in this paper, we adopt single-image super-resolution (SR) as a representative testbed and employ SwinIR (Liang et al., 2021) to explore perceptual optimization under diverse fidelity metrics and discriminator designs, with particular emphasis on their relationship to perceptual assessment. To this end, we first construct DISTS-style (Ding et al., 2020) *injective* perceptual metrics with varied visual backbones, thereby spanning a range of objectives with different IQA capacities, and instantiate four distinct optimization objective settings in practice for optimizing the SwinIR model. We then design discriminators based on both convolutional and Transformer architectures, and investigate the transferability of their learned representations by initializing IQA models with backbones extracted from SR-trained discriminators, in comparison to ImageNet-pretrained (Russakovsky et al., 2015) and randomly initialized counterparts under both FR and no-reference (NR) settings. Finally, given the central role of discriminators in enforcing

perceptual realism and mitigating artifacts, we extend our analysis to discriminator variants, focusing on the widely used vanilla and patch-level designs[1], and evaluate their training stability across alternative architectures, thereby advancing a deeper understanding of their capacity in facilitating more effective perceptual optimization. Collectively, our study leads to the following key findings:

- We investigate the link between IQA-oriented fidelity metrics and optimization, and reveal that higher IQA capability does not necessarily translate into more effective optimization guidance, particularly when adversarial objectives are involved.

- We analyze the representational transfer of discriminators and demonstrate that, despite their effectiveness in artifact suppression, their learned features contribute little when repurposed for IQA initialization, highlighting a fundamental asymmetry between optimization and assessment roles.

- We conduct a controlled investigation of discriminator design, showing that patch-level discriminators enable more faithful detail reconstruction than their vanilla counterparts, while convolutional-based discriminators demonstrate superior training stability relative to Transformer-based alternatives.

## 2 FORMULATION OF PERCEPTUAL OPTIMIZATION OBJECTIVES

In this section, we elaborate on the formulation of perceptual optimization losses, comprising composite, DISTS-style perceptual, and adversarial components.

### 2.1 COMPOSITE OBJECTIVE FOR PERCEPTUAL OPTIMIZATION

Perceptual optimization typically employs a composite objective that integrates two fidelity terms, reconstruction and perceptual, together with an adversarial objective (Wang et al., 2018; Sun et al., 2024). The reconstruction loss $\ell_{\text{rec}}$ enforces fidelity in low-frequency structures and color consistency (Dong et al., 2015), while the perceptual loss $\ell_{\text{per}}$ measures discrepancies in deep feature space and thereby encourages structural alignment and fine-grained texture preservation in accordance with the HVS (Zhang et al., 2018; Ding et al., 2020). Complementing these, the adversarial loss $\ell_{\text{adv}}$ promotes realism by pushing generated outputs toward the distribution of natural images (Goodfellow et al., 2014). Formally, the overall objective can be expressed as:

$$\ell(x, y) = \lambda_1 \ell_{\text{rec}}(x, y) + \lambda_2 \ell_{\text{per}}(x, y) + \lambda_3 \ell_{\text{adv}}(x, y), \tag{1}$$

where $x \in \mathcal{X}$ and $y \in \mathcal{Y}$ denote the generated and reference images, respectively, and $\lambda_1, \lambda_2, \lambda_3 \geq 0$ are balancing weights. In this paper, we particularly focus on the latter two components, $\ell_{\text{per}}$ and $\ell_{\text{adv}}$, as they play decisive roles in shaping perceptual realism. Our analysis therefore centers on disentangling their respective contributions and interactions, with the goal of clarifying their influence on optimization outcomes and their potential transferability to IQA tasks.

### 2.2 COMPUTATION OF DISTS-STYLE PERCEPTUAL METRICS

To rigorously investigate the extent to which IQA performance reflects optimization utility, we develop a family of perceptual metrics derived from the DISTS framework (Ding et al., 2020), wherein an *injective* feature transformation is applied to capture deep representations and global similarity is assessed across texture and structure dimensions. The symmetric perceptual metric $\ell_{\text{per}}(x, y)$ is defined by first applying a visual feature transformation to obtain deep representations of $x$ and $y$. Let $x_j^{(i)}$ and $y_j^{(i)}$ denote the $j$-th channel of the $i$-th layer features extracted from $x$ and $y$, respectively. The texture similarity $l(\cdot)$ and structure similarity $s(\cdot)$ are computed as:

$$l\left(x_j^{(i)}, y_j^{(i)}\right) = \frac{2\mu_{x_j}^{(i)}\mu_{y_j}^{(i)} + c_1}{\left(\mu_{x_j}^{(i)}\right)^2 + \left(\mu_{y_j}^{(i)}\right)^2 + c_1}, \; s\left(x_j^{(i)}, y_j^{(i)}\right) = \frac{2\sigma_{x_j y_j}^{(i)} + c_2}{\left(\sigma_{x_j}^{(i)}\right)^2 + \left(\sigma_{y_j}^{(i)}\right)^2 + c_2}, \tag{2}$$

---

[1]The vanilla discriminator outputs a single global score for the entire image, whereas the patch-level discriminator generates a map of real/fake predictions across local patches.

where $\mu_{x_j}^{(i)}$ and $\mu_{y_j}^{(i)}$ are the global means of $x_j^{(i)}$ and $y_j^{(i)}$, $\sigma_{x_j}^{(i)}$ and $\sigma_{y_j}^{(i)}$ are their variances, and $\sigma_{x_j y_j}^{(i)}$ denotes the global covariance. Constants $c_1$ and $c_2$ are introduced to avoid numerical instability when denominators are close to zero. The overall DISTS-style perceptual metric is then formulated as a weighted sum of texture and structure similarities across all feature layers and channels:

$$\ell_{\text{per}}(x, y) = 1 - \sum_{i=0}^{m} \sum_{j=1}^{n_i} \left( \alpha_{ij} \cdot l\left(x_j^{(i)}, y_j^{(i)}\right) + \beta_{ij} \cdot s\left(x_j^{(i)}, y_j^{(i)}\right) \right), \tag{3}$$

where $m$ is the total number of feature layers, $n_i$ represents the number of channels in the $i$-th layer, and $\alpha_{ij}, \beta_{ij} \geq 0$ denote learnable parameters that are required to satisfy the normalization constraint $\sum_{i=0}^{m} \sum_{j=1}^{n_i} (\alpha_{ij} + \beta_{ij}) = 1$.

## 2.3 ADVERSARIAL LOSS IN PERCEPTUAL OPTIMIZATION

Adversarial loss has become a cornerstone in visual generation tasks (Ledig et al., 2017; Esser et al., 2021; Sauer et al., 2024), as it enhances perceptual realism by complementing fidelity objectives within perceptual optimization frameworks. It is formulated as a two-player game between a generator and a discriminator, where the generator strives to synthesize realistic images while the discriminator learns to differentiate them from real samples. This adversarial interplay drives the generator toward outputs that are not only structurally faithful but also perceptually convincing.

Within existing adversarial formulations, relativistic variants of GANs (Jolicoeur-Martineau, 2018) have been shown to be particularly effective for perceptual optimization. Following ESR-GAN (Wang et al., 2018), we adopt an adversarial loss design based on a relativistic discriminator. The generator loss is expressed as:

$$\ell_{\text{adv}}(x, y) = -\mathbb{E}_{y \sim \mathcal{Y}} \left[ \log \left( 1 - D(y, x) \right) \right] - \mathbb{E}_{x \sim \mathcal{X}} \left[ \log \left( D(x, y) \right) \right], \tag{4}$$

while the discriminator loss is given by

$$\ell_d(x, y) = -\mathbb{E}_{y \sim \mathcal{Y}} \left[ \log \left( D(y, x) \right) \right] - \mathbb{E}_{x \sim \mathcal{X}} \left[ \log \left( 1 - D(x, y) \right) \right], \tag{5}$$

where the asymmetric discrepancy $D(x, y)$ is defined as:

$$D(x, y) = \sigma \left( d(x) - \mathbb{E}_{y \sim \mathcal{Y}} \left[ d(y) \right] \right), \tag{6}$$

with $\sigma(\cdot)$ denoting the sigmoid function, $d(\cdot)$ the discriminator, and $\mathbb{E}[\cdot]$ the mini-batch average.

## 3 EXPERIMENTS

In this section, we systematically investigate perceptual optimization across diverse fidelity metrics and discriminator designs, with emphasis on their relation to quality assessment and on the stability and effectiveness of alternative discriminator architectures.

### 3.1 EXPERIMENTAL SETUPS

**Construction of Perceptual Metrics** We construct a family of DISTS-style perceptual metrics (Ding et al., 2020) by replacing the visual backbone introduced in Sec. 2.2, yielding objectives with different levels of IQA capability. Specifically, we examine three convolutional architectures: VGG-16 (Simonyan & Zisserman, 2014), ResNet-50 (He et al., 2016), and ConvNeXt (Liu et al., 2022), as well as two Transformer-based architectures, CLIP-ViT (Radford et al., 2021) and Swin Transformer (Liu et al., 2021). We also include a variant that employs a VGG-16 backbone, in which both the network parameters and the weighting factors $\alpha_{ij}$ and $\beta_{ij}$ are randomly assigned and kept fixed (denoted VGG-16-R). Together, these configurations yield the perceptual objectives $\ell_{\text{VGG-16}}$, $\ell_{\text{ResNet-50}}$, $\ell_{\text{ConvNeXt}}$, $\ell_{\text{CLIP-ViT}}$, $\ell_{\text{Swin-T}}$, and $\ell_{\text{VGG-16-R}}$. For evaluation, we adopt four synthetic FR benchmarks: TID2013 (traditional) (Ponomarenko et al., 2015), Liu13 (deblurring) (Liu et al., 2013), Ma17 (super-resolution) (Ma et al., 2017), and TQD (texture similarity) (Ding et al., 2020), to comprehensively assess IQA performance across diverse scenarios. The complete training protocol and results are provided in the Appendix.

Table 1: **Comparison of perceptual metrics** on the DIV2K validation set. The final two columns present the $\ell_1$-only and reference results. "Std." denotes the standard deviation of NR-IQA scores. Top two results are highlighted in **bold** and underline, respectively.

| Metric | VGG-16-R | VGG-16 | ResNet-50 | ConvNeXt | CLIP-ViT | Swin-T | Std. | $\ell_1$ | Ref. |
|---|---|---|---|---|---|---|---|---|---|
| *Perceptual* | | | | | | | | | |
| MANIQA | 49.98 | 54.03 | 46.66 | **54.50** | 28.02 | 50.31 | 8.995 | 48.50 | 61.78 |
| LIQE | 58.90 | 58.30 | 54.40 | **62.46** | 59.02 | 59.30 | 2.355 | 54.57 | 67.50 |
| DeQA | **64.05** | 64.04 | 60.71 | 54.05 | 41.56 | 61.62 | 7.945 | 53.56 | 64.96 |
| VQ-R1 | 59.04 | **61.12** | 55.72 | 55.09 | 39.56 | 60.16 | 7.294 | 51.31 | 63.21 |
| Average | 57.99 | **59.37** | 54.37 | 56.53 | 42.04 | 57.85 | 5.863 | 51.98 | 64.36 |
| *Reconstruction and Perceptual* | | | | | | | | | |
| MANIQA | 50.05 | 53.96 | 48.85 | **54.27** | 38.67 | 50.28 | 5.182 | 48.50 | 61.78 |
| LIQE | 58.91 | 58.39 | 56.53 | 61.92 | **66.11** | 59.74 | 3.068 | 54.57 | 67.50 |
| DeQA | **63.35** | 62.36 | 62.60 | 60.49 | 54.58 | 54.31 | 3.756 | 53.56 | 64.96 |
| VQ-R1 | 59.32 | **60.59** | 58.95 | 59.95 | 52.07 | 56.07 | 2.940 | 51.31 | 63.21 |
| Average | 57.91 | 58.83 | 56.73 | **59.16** | 52.86 | 55.10 | 2.211 | 51.98 | 64.36 |
| *Perceptual and Adversarial* | | | | | | | | | |
| MANIQA | 45.97 | **57.28** | 56.42 | 56.66 | 56.95 | 56.83 | 4.055 | 48.50 | 61.78 |
| LIQE | 65.53 | 66.76 | 67.66 | 67.20 | **67.72** | 67.62 | 0.770 | 54.57 | 67.50 |
| DeQA | 55.95 | 62.04 | **62.57** | 61.54 | 61.83 | 61.98 | 2.273 | 53.56 | 64.96 |
| VQ-R1 | 55.37 | 61.13 | **61.82** | 61.15 | 61.44 | 61.06 | 2.232 | 51.31 | 63.21 |
| Average | 55.71 | 61.80 | **62.12** | 61.64 | 61.99 | 61.87 | 2.307 | 51.98 | 64.36 |
| *Reconstruction, Perceptual and Adversarial* | | | | | | | | | |
| MANIQA | 48.04 | 56.50 | **57.73** | 56.21 | 55.33 | 56.90 | 3.247 | 48.50 | 61.78 |
| LIQE | 66.93 | 67.87 | 67.82 | 66.73 | 67.37 | **67.90** | 0.467 | 54.57 | 67.50 |
| DeQA | 59.32 | 62.59 | **62.75** | 61.06 | 60.01 | 62.36 | 1.324 | 53.56 | 64.96 |
| VQ-R1 | 57.15 | 61.41 | **61.81** | 60.42 | 60.45 | 61.04 | 1.526 | 51.31 | 63.21 |
| Average | 57.86 | 62.09 | **62.53** | 61.11 | 60.79 | 62.05 | 1.555 | 51.98 | 64.36 |

**Configuration of SR Model Training**  We conduct perceptual optimization for SR using SwinIR (Liang et al., 2021), a widely adopted discriminative backbone. Training is performed in two stages. In the first stage, the model is trained for 100K iterations with an $\ell_1$ reconstruction loss and an initial learning rate of $2 \times 10^{-4}$. In the second stage, starting from weights pre-trained with reconstruction loss, we evaluate four settings: (1) perceptual loss only, (2) combined reconstruction and perceptual losses, (3) perceptual plus adversarial loss, and (4) the full objective in Eq. 1, where the weights are fixed to $\lambda_1 = 1 \times 10^{-2}$, $\lambda_2 = 1$, and $\lambda_3 = 5 \times 10^{-3}$ as in (Wang et al., 2018). For settings (3) and (4), the discriminator is a vanilla VGG-16 (Simonyan & Zisserman, 2014). This stage runs for 400K iterations with the same initial learning rate, decayed by half at 150K, 300K, 350K, and 375K steps. For adversarial objectives, generator and discriminator are alternately updated. All experiments are carried out on the DIV2K dataset (Agustsson & Timofte, 2017), where low-resolution inputs are generated by applying $4\times$ bicubic downsampling to $256 \times 256$ reference images. We adopt Adam (Kingma & Ba, 2014) with batch size 32, and train all models in PyTorch on NVIDIA L40S GPUs.

**Metrics for Evaluating Visual Quality**  To rigorously evaluate the visual quality of images produced by models trained with different optimization objectives, we adopt four state-of-the-art NR-IQA methods[2]: MANIQA (Yang et al., 2022), LIQE (Zhang et al., 2023), DeQA-Score (DeQA) (You et al., 2025), and VisualQuality-R1 (VQ-R1) (Wu et al., 2025), and normalize their outputs following Wu et al. (2024b); Chen et al. (2025) (Details are shown in the Appendix).

## 3.2 ANALYSIS OF PERCEPTUAL METRICS

Table 4 and Table 1 comprehensively summarize the performance of various perceptual metrics, reporting both their IQA accuracy across multiple benchmarks and their effectiveness in guiding perceptual optimization for SR training. From these results, several insightful observations can be drawn, shedding light on the interplay between evaluation capability and optimization efficacy.

**Misalignment between Evaluation and Optimization**  Strong IQA performance does not reliably confer effective optimization guidance. As shown in Fig. 3, IQA capability exhibits no consistent

---

[2]FR-IQA metrics are unsuitable in this setting, as SR models optimized with distinct loss functions (*e.g.*, models trained for PSNR are inherently biased toward PSNR-based evaluations) may yield unfair comparisons.

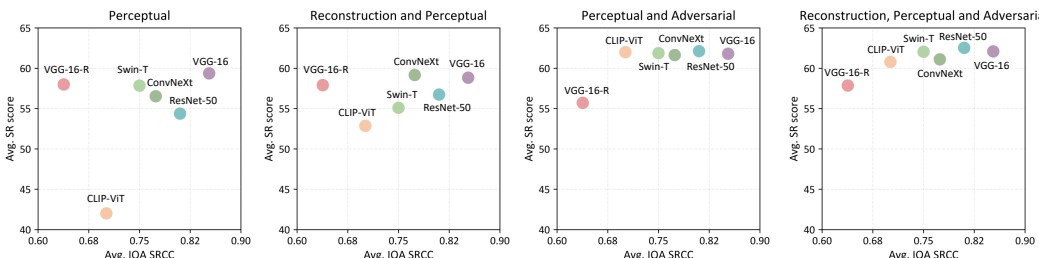

Figure 3: **Correlation** between average IQA SRCC values and average SR visual quality scores. Across all settings, higher IQA SRCC does not reliably yield better perceptual optimization; the association is weak at best and becomes especially tenuous when adversarial loss is involved.

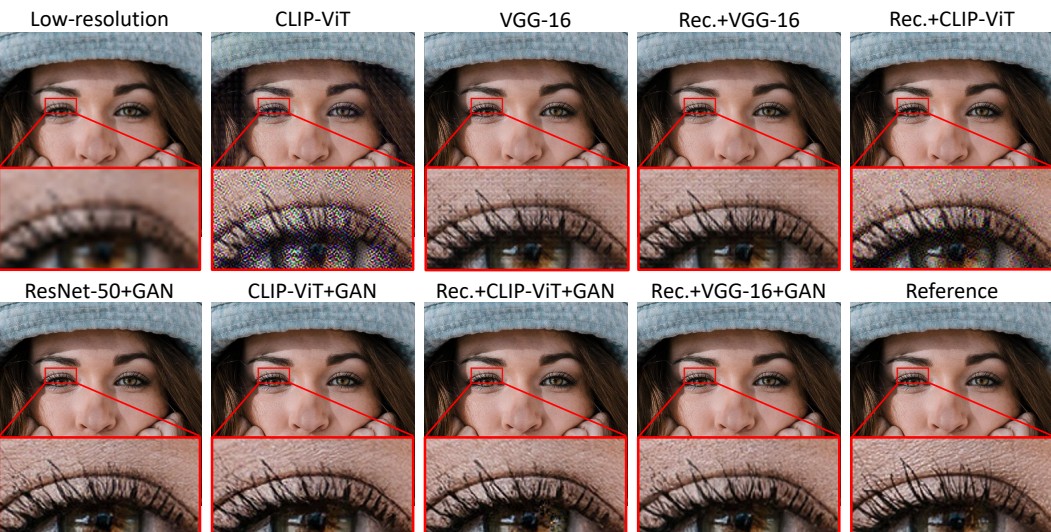

Figure 4: **Qualitative comparison** of SR results under different optimization objectives. In the second training stage, adding a reconstruction penalty yields negligible improvements; by contrast, adversarial loss substantially mitigates checkerboard artifacts. Moreover, with adversarial loss, SR outputs optimized by different perceptual metrics show only minor visual differences.

correlation with optimization outcomes across the four training configurations. A striking example is the randomly initialized $\ell_{\text{VGG-16-R}}$, which, despite ranking last on all FR-IQA benchmarks, still significantly surpasses $\ell_{\text{ResNet-50}}$ and $\ell_{\text{CLIP-ViT}}$ in both the perceptual-only and reconstruction-plus-perceptual settings. This discrepancy underscores a fundamental mismatch between the roles of perceptual metrics when used as evaluators and when employed as optimization objectives.

**Marginal Role of Reconstruction Supervision**  As shown in Table 1, incorporating a reconstruction term yields only marginal improvements, irrespective of adversarial supervision, implying a redundancy of pixel-level guidance in second-stage perceptual optimization. In line with this observation, Fig. 4 demonstrates that the combined objective "Rec.+VGG-16" offers negligible advantage over $\ell_{\text{VGG-16}}$ alone in mitigating checkerboard artifacts, further underscoring the limited contribution of pixel-level supervision in this context.

**Effectiveness of Adversarial Supervision**  In contrast to the reconstruction term, adversarial loss $\ell_{\text{adv}}$ generally improves optimization across metrics, except for $\ell_{\text{VGG-16-R}}$. As shown in Fig. 4, it yields sharper textures, more realistic details, and fewer artifacts than non-adversarial counterparts. These results suggest that, when combined with a fidelity metric of sufficient IQA capability, adversarial supervision not only mitigates artifact formation but also complements perceptual metrics by steering SR training toward closer alignment with human visual preferences.

**Homogenization Effect of Adversarial Supervision**  The standard deviation values in Table 1 show that adversarial supervision markedly reduces disparities across perceptual metrics: in the

Table 2: **Quantitative comparison** of FR-IQA performance across FR benchmarks using VGG-16, DINOv2, and ResNet-50 with random, GAN, and ImageNet initializations.

| Backbone | TID2013 | | Liu13 | | Ma17 | | TQD | | Average | |
|---|---|---|---|---|---|---|---|---|---|---|
| | SRCC | PLCC | SRCC | PLCC | SRCC | PLCC | SRCC | PLCC | SRCC | PLCC |
| *VGG-16* | | | | | | | | | | |
| Random | 0.782 | 0.793 | 0.860 | 0.873 | 0.750 | 0.758 | 0.374 | 0.549 | 0.692 | 0.743 |
| GAN | 0.727 | 0.753 | 0.859 | 0.862 | 0.805 | 0.829 | 0.303 | 0.466 | 0.674 | 0.727 |
| ImageNet | **0.873** | **0.895** | **0.929** | **0.935** | **0.895** | **0.908** | **0.715** | **0.731** | **0.853** | **0.867** |
| *ResNet-50* | | | | | | | | | | |
| Random | 0.745 | 0.751 | 0.808 | 0.839 | 0.737 | 0.744 | 0.379 | 0.521 | 0.667 | 0.714 |
| GAN | 0.760 | 0.775 | 0.823 | 0.816 | 0.752 | 0.786 | 0.312 | 0.496 | 0.662 | 0.718 |
| ImageNet | **0.871** | **0.896** | **0.900** | **0.912** | **0.866** | **0.886** | **0.604** | **0.680** | **0.810** | **0.843** |
| *DINOv2* | | | | | | | | | | |
| Random | 0.813 | 0.817 | 0.839 | 0.856 | 0.753 | 0.754 | **0.429** | **0.548** | 0.708 | 0.744 |
| GAN | 0.767 | 0.791 | 0.851 | 0.870 | 0.759 | 0.773 | 0.361 | 0.511 | 0.685 | 0.736 |
| ImageNet | **0.816** | **0.860** | **0.913** | **0.920** | **0.897** | **0.910** | 0.276 | 0.531 | **0.726** | **0.805** |

Figure 5: **Quantitative comparison** of NR-IQA results on KADID-10K with VGG-16, DINOv2, and ResNet-50 under random, GAN, and ImageNet initialization.

perceptual-only setting the deviation reaches 5.863, whereas with the full objective in Eq. 1 it drops to 1.555. Fig. 3 further corroborates this homogenization, as the inclusion of $\ell_{\mathrm{adv}}$ yields tighter clustering of metrics in SR average score, thereby weakening the link between IQA accuracy and optimization effectiveness. Consistent with this trend, Fig. 4 shows that visual differences among methods become markedly less discernible, with reconstructed outputs appearing increasingly similar in texture and structure. Collectively, these results suggest that once a reasonably accurate perceptual metric is in place, adversarial loss tends to exert a dominant influence on the optimization process, raising the critical question of whether further advances in perceptual metric design can remain impactful in the presence of adversarial training.

### 3.3 GAN-BASED INITIALIZATION FOR IQA

Building on the observation that adversarial supervision mitigates artifacts and enhances perceptual realism in SR, we investigate whether discriminator-learned representations encode perceptually relevant cues transferable to IQA models. We employ the convolutional architectures VGG-16 (Simonyan & Zisserman, 2014) and ResNet-50 (He et al., 2016), together with the Transformer architecture DINOv2 (Oquab et al., 2023)[3], for SR adversarial training. The resulting discriminator backbones are then used as initializations for both FR- and NR-IQA models, and their effectiveness is compared against random initialization and ImageNet pretraining (Russakovsky et al., 2015). For evaluation, we assess FR-IQA models on four synthetic benchmarks described in Sec. 3.1, and evaluate NR-IQA models on the KADID (Lin et al., 2019) test set. The training configurations for both FR and NR settings are detailed in the Appendix.

**Limited Transferability of Discriminator Features** Table 2 and Fig. 5 indicate that ImageNet pretraining is consistently the most effective initialization, producing the highest SRCC and PLCC across all backbones and datasets. In comparison, initializing with GAN-trained discriminators yields only small gains over random initialization for ResNet-50 and DINOv2, with especially modest improvements for DINOv2. These findings suggest that adversarial supervision captures cues

---

[3]All networks are randomly initialized without loading any pretrained weights.

| Discriminator | MANIQA | LIQE | DeQA | VQ-R1 | Average |
|---|---|---|---|---|---|
| *Vanilla* | | | | | |
| VGG-16 | 56.50 | 67.87 | 62.59 | 61.41 | 62.09 |
| ResNet-50 | **59.01** | 68.03 | 64.81 | 62.29 | 63.54 |
| DINOv2 | 57.81 | 67.20 | 62.48 | 61.23 | 62.18 |
| *Patch-level* | | | | | |
| VGG-16 | 57.74 | 67.72 | 63.20 | 61.79 | 62.61 |
| ResNet-50 | 58.87 | **68.24** | **66.05** | **62.51** | **63.92** |
| DINOv2 | 57.79 | 67.57 | 62.74 | 61.22 | 62.33 |
| w/o (base) | 53.96 | 58.39 | 62.36 | 60.59 | 58.83 |

Table 3: **Quantitative comparison** of discriminator architectures in adversarial perceptual optimization. We evaluate vanilla and patch-level discriminators with VGG-16, ResNet-50, and DINOv2 backbones across four NR-IQA metrics, providing a systematic assessment of their relative effectiveness in guiding perceptual optimization.

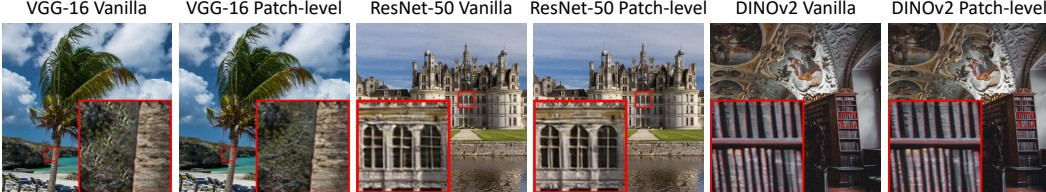

VGG-16 Vanilla  VGG-16 Patch-level  ResNet-50 Vanilla  ResNet-50 Patch-level  DINOv2 Vanilla  DINOv2 Patch-level

Figure 6: **Qualitative comparison** of vanilla vs. patch-level discriminators.

related to artifact suppression, but the resulting representations lack the semantic coverage and robustness required for generalizable quality assessment. This gap likely arises from a mismatch between the discriminator's narrow real-fake discrimination objective and IQA's broader requirement for sensitivity to diverse distortions and content.

## 3.4 FURTHER ANALYSIS OF DISCRIMINATORS

The incorporation of adversarial loss has been shown to significantly enhance optimization toward human visual preference by facilitating the recovery of high-frequency details and perceptually plausible structures. Nonetheless, the effectiveness and stability of adversarial training are highly dependent on the design of the discriminator. To investigate this, we conduct a systematic comparison between two widely adopted discriminator architectures: **vanilla** and **patch-level**. In our experiments, we adopt VGG-16 (Simonyan & Zisserman, 2014), ResNet-50 (He et al., 2016), and DINOv2 (Oquab et al., 2023) as discriminator backbones, modifying only the regression head to produce either vanilla or patch-level outputs, while employing $\ell_1$ and $\ell_{\text{VGG-16}}$ as the reconstruction and perceptual objectives, respectively.

**Analysis of Effectiveness** Table 3 reports a comparison between vanilla and patch-level discriminators for adversarial supervision in perceptual optimization. Results show that introducing any discriminator markedly improves performance over the non-adversarial baseline. For VGG-16 and ResNet-50 backbones, patch-level designs consistently surpass vanilla ones, raising the average score by +0.52 and +0.38 points, respectively. The qualitative results in Fig. 6 align with these findings: patch-level discriminators generate sharper textures, clearer local structures, and fewer artifacts. By contrast, the DINOv2 backbone exhibits only a marginal gain (+0.15 average),

suggesting that Transformer-based discriminators are less responsive to patch-level supervision in this setting. Overall, patch-level supervision emerges as a more effective and reliable default choice for convolutional backbones, delivering enhanced robustness and stronger guidance for local detail preservation.

**Analysis of Training Stability** Generative adversarial training is notoriously fragile, being prone to non-convergence and mode collapse, and highly sensitive to hyperparameters (Mescheder et al., 2018). To probe this aspect, we examine training stability of patch-level discriminators under varying adversarial weights $\lambda_3$ in Eq. 1 across different discriminator backbones. Fig. 7 shows

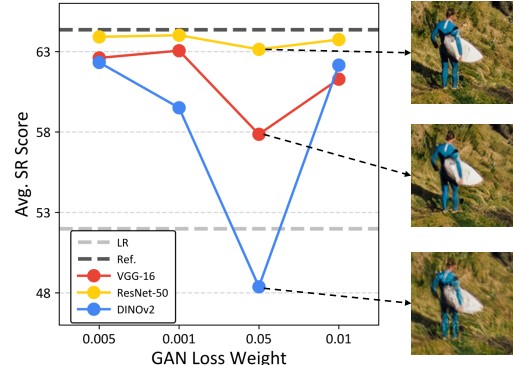

Figure 7: **Impact** of GAN loss weight.

that the ResNet-50 discriminator achieves consistently high SR scores across a broad range of weights, indicating robustness to hyperparameter variation, whereas VGG-16 shows moderate sensitivity with performance dropping at larger weights. By contrast, DINOv2 suffers from severe instability, with performance collapsing when the adversarial weight is increased, underscoring its limited suitability for stable adversarial optimization. This suggests that convolutional discriminators are inherently more stable, likely due to their stronger inductive biases and locality modeling.

## 4 RELATED WORK

**Perceptual Optimization**   Perceptual optimization has become a central paradigm in image restoration and generation, motivated by the limitations of pixel-wise fidelity objectives such as $\ell_1$ norm or PSNR, which often fail to reflect human perceptual judgments. Early works introduced perceptual losses based on deep features from pretrained networks like VGG (Johnson et al., 2016; Ledig et al., 2017), shifting the focus from pixel correspondence to semantic and structural fidelity. Subsequent advances proposed task-specific perceptual metrics (Zhang et al., 2018; Chen et al., 2024; Lao et al., 2022) that better align with human opinion scores. In parallel, adversarial training emerged as a complementary objective, with discriminators encouraging sharper textures and more perceptually realistic local structures. This paradigm has since been applied across diverse restoration and generation tasks (Wang et al., 2018; Sun et al., 2024; Tian et al., 2024). While prior studies (Blau & Michaeli, 2018; Ding et al., 2020; 2021b) mainly compared perceptual metrics for optimization, our work systematically examines their interplay with evaluation, with particular emphasis on the role of adversarial discriminators, thereby addressing an important gap.

**Perceptual Metrics**   Perceptual metrics aim to quantify visual differences between a test image and its reference. Conventional FR-IQA methods assume the reference is of perfect quality, so the measured distance directly reflects test image quality. Early error-based measures such as MSE and MAE dominated the field but often diverge from human perception. To address this, SSIM (Wang et al., 2004) emphasized structural fidelity, later extended to multiscale (Wang et al., 2003) and feature-domain forms (Zhang et al., 2011). Recent deep-feature approaches, including DISTS (Ding et al., 2020) and its adaptive variant (Ding et al., 2021a), better capture structure and texture. Moving beyond the perfect-reference assumption, asymmetric metrics such as VIF (Sheikh & Bovik, 2006), CKDN (Zheng et al., 2021), and A-FINE (Chen et al., 2025) allow test images to surpass the reference. In perceptual optimization, however, most metrics still guide models to approximate the given reference, implicitly treating it as the perceptual upper bound.

## 5 CONCLUSION AND DISCUSSION

We systematically examined how perceptual metrics and discriminator architectures affect perceptual optimization and IQA. We find that stronger fidelity metrics do not guarantee better optimization, discriminator features transfer poorly to IQA, and patch-level convolutional discriminators yield more stable and detailed results than vanilla or Transformer-based ones.

**Limitations and Future Directions**   This study has several limitations that suggest directions for future work. First, our experiments were conducted exclusively on SwinIR (Liang et al., 2021); extending the analysis to relatively weaker models (*e.g.*, SRResNet (Ledig et al., 2017)), stronger models (*e.g.*, HAT (Chen et al., 2023)), or even training visual tokenizers (Sun et al., 2024) would provide a more comprehensive test of robustness. Second, the limited transferability of discriminator features to IQA may partly stem from the relatively small and narrow SR training data compared with large-scale, diverse datasets such as ImageNet (Russakovsky et al., 2015); scaling up training with more diverse data would allow for a fairer and more conclusive assessment. Third, our study also reveals the pronounced role of discriminators in the second-stage SR training. Building on this, incorporating discriminator-driven designs into post-training for image generation with reinforcement learning (Liu et al., 2025) appears to be a promising direction for future exploration. Finally, by highlighting this asymmetry, we question the prevailing practice of relying solely on perceptual metrics for optimization and assessing their capability based on the resulting outcomes (Ding et al., 2021b). We hope these findings motivate broader and sustained efforts to co-design perceptual metrics, adversarial objectives, and evaluation protocols, thereby ultimately advancing more principled, robust, and generalizable perceptual modeling.

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

APPENDIX

This appendix provides details on the use of large language models (LLMs), training configurations, metric rescaling methods, additional results, and qualitative visualizations.

## A THE USE OF LARGE LANGUAGE MODELS

LLMs were utilized exclusively for refining the manuscript and assisting with coding. They were not employed for idea generation, literature retrieval, or any other aspects of the research process.

## B TRAINING CONFIGURATIONS FOR DISTS-STYLE PERCEPTUAL METRICS

In perceptual metric training, the backbone is fixed to preserve generalizability (Ding et al., 2020; Chen et al., 2025). We use Adam (Kingma & Ba, 2014) with an initial learning rate of $1 \times 10^{-4}$, halved every 1,000 iterations. Following Ding et al. (2020), the zeroth-stage weights are projected onto $[0.02, 1]$ after each update for stability. All models are trained for 5,000 iterations on KADID-10K (Lin et al., 2019) with a batch size of 32.

## C TRAINING CONFIGURATIONS IN GAN-BASED INITIALIZATION

For FR-IQA model training, VGG-16 (Simonyan & Zisserman, 2014), ResNet-50 (He et al., 2016), and DINOv2 (Oquab et al., 2023) are trained using the same strategies described in Sec. B. For NR-IQA model training, we conduct experiments on KADID-10K (Lin et al., 2019) and KonIQ-10k (Hosu et al., 2020), adopting a 6:2:2 division into training, validation, and testing sets. Model performance is reported on the test set using SRCC and PLCC.

## D RESCALING VISUAL QUALITY METRICS

To facilitate a fair and interpretable comparison of perceptual quality across models, we follow Ding et al. (2020); Chen et al. (2025); Wu et al. (2024b) and normalize the outputs of these NR-IQA models onto a unified perceptual scale ranging from 1 to 100 by means of a four-parameter monotonic logistic mapping:

$$N_\eta(x) = \frac{\eta_1 - \eta_2}{1 + \exp\left(-\frac{N(x)-\eta_3}{|\eta_4|}\right)} + \eta_2, \tag{7}$$

where $N(x)$ denotes the raw score predicted by an NR-IQA model. The parameters $\eta_1$ and $\eta_2$ are fixed to 100 and 1, respectively, defining the upper and lower bounds of the normalized scale. The remaining parameters, $\eta_3$ and $\eta_4$, are estimated during fitting. Under this mapping, larger normalized values correspond to higher perceived image quality.

## E ADDITIONAL RESULTS

**Results of DISTS-style Perceptual Metrics**  The results in Table 4 demonstrate clear differences in performance across backbones on the four synthetic FR-IQA datasets. $\ell_{\text{VGG-16}}$ achieves the highest overall performance, with average SRCC and PLCC values of 0.853 and 0.867, respectively, consistently ranking first across datasets. $\ell_{\text{ResNet-50}}$ follows closely, reaching the best PLCC on TID2013 (0.896) and yielding strong average performance (0.810/0.843), ranking second overall. $\ell_{\text{ConvNeXt}}$ and $\ell_{\text{Swin-T}}$ achieve intermediate results. By contrast, $\ell_{\text{CLIP-ViT}}$ and $\ell_{\text{VGG-16-R}}$ lag behind, leading to the lowest overall rank. Overall, convolutional backbones substantially outperform Transformer-based ones, suggesting that convolutional features remain better aligned with perceptual quality assessment under synthetic distortions.

**Quantitative Comparison of Backbone Initializations**  Consistent with the findings in Sec. 3.3, the results in Fig. 8 show that ImageNet pretraining (Russakovsky et al., 2015) yields the best NR-IQA performance across all backbones, underscoring the importance of large-scale supervised ini-

Table 4: **Quantitative Comparison** of SRCC and PLCC across perceptual metrics on four synthetic FR IQA datasets.

| Backbone | TID2013 | | Liu13 | | Ma17 | | TQD | | Average | | Rank |
|---|---|---|---|---|---|---|---|---|---|---|---|
| | SRCC | PLCC | SRCC | PLCC | SRCC | PLCC | SRCC | PLCC | SRCC | PLCC | |
| VGG-16-R | 0.643 | 0.651 | 0.799 | 0.822 | 0.818 | 0.815 | 0.293 | 0.493 | 0.638 | 0.695 | 6 |
| VGG-16 | **0.873** | 0.895 | **0.929** | **0.935** | **0.895** | **0.908** | **0.715** | **0.731** | **0.853** | **0.867** | 1 |
| ResNet-50 | 0.871 | **0.896** | 0.900 | 0.912 | 0.866 | 0.886 | 0.604 | 0.680 | 0.810 | 0.843 | 2 |
| ConvNeXt | 0.780 | 0.833 | 0.884 | 0.900 | 0.879 | 0.893 | 0.553 | 0.643 | 0.774 | 0.817 | 3 |
| CLIP-ViT | 0.808 | 0.858 | 0.912 | 0.919 | 0.790 | 0.835 | 0.293 | 0.472 | 0.701 | 0.771 | 5 |
| Swin-T | 0.708 | 0.799 | 0.856 | 0.886 | 0.848 | 0.860 | 0.587 | 0.656 | 0.750 | 0.800 | 4 |

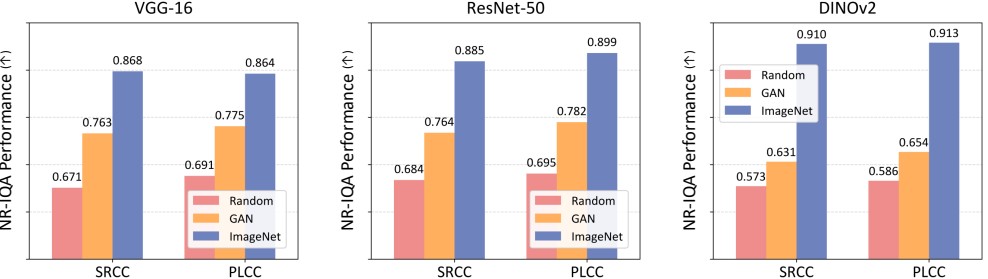

Figure 8: **Quantitative comparison** of NR-IQA results on KonIQ-10k with VGG-16, DINOv2, and ResNet-50 under random, GAN, and ImageNet initialization.

tialization. GAN-based initialization provides clear improvements over random initialization, suggesting that adversarial training captures perceptually relevant cues, though its effectiveness remains limited compared to ImageNet. Among backbones, DINOv2 achieves the highest scores under pretraining but performs poorly with random or GAN initialization, indicating that Transformer architectures are more reliant on large-scale pretraining, whereas convolutional networks maintain moderate robustness under weaker initializations.

# F  ADDITIONAL VISUALIZATIONS

We provide more visualizations of different optimization settings illustrated in Sec. 3.1, shown in Fig. 9, Fig. 10, and Fig. 11. Across images, pure reconstruction produces smooth yet over-smoothed results, with visible zippering or checkerboard artifacts for VGG-16-R and VGG-16. Perceptual loss improves edge continuity and global structure, but ViT backbones tend to hallucinate elongated streaks in fine textures. Adding an adversarial term recovers stochastic details in the feathers and rock granularity while suppressing grid artifacts. Convolutional discriminators paired with CNN backbones (ResNet-50 (He et al., 2016), ConvNeXt (Liu et al., 2022), VGG-16 (Simonyan & Zisserman, 2014)) yield the most coherent local textures and consistent contrast, whereas Transformer backbones (CLIP-ViT (Radford et al., 2021), Swin-T (Liu et al., 2021)) remain more prone to banding and unstable micro-patterns. The combined setting with a reconstruction term plus perceptual loss and GAN provides the best balance between fidelity and realism.

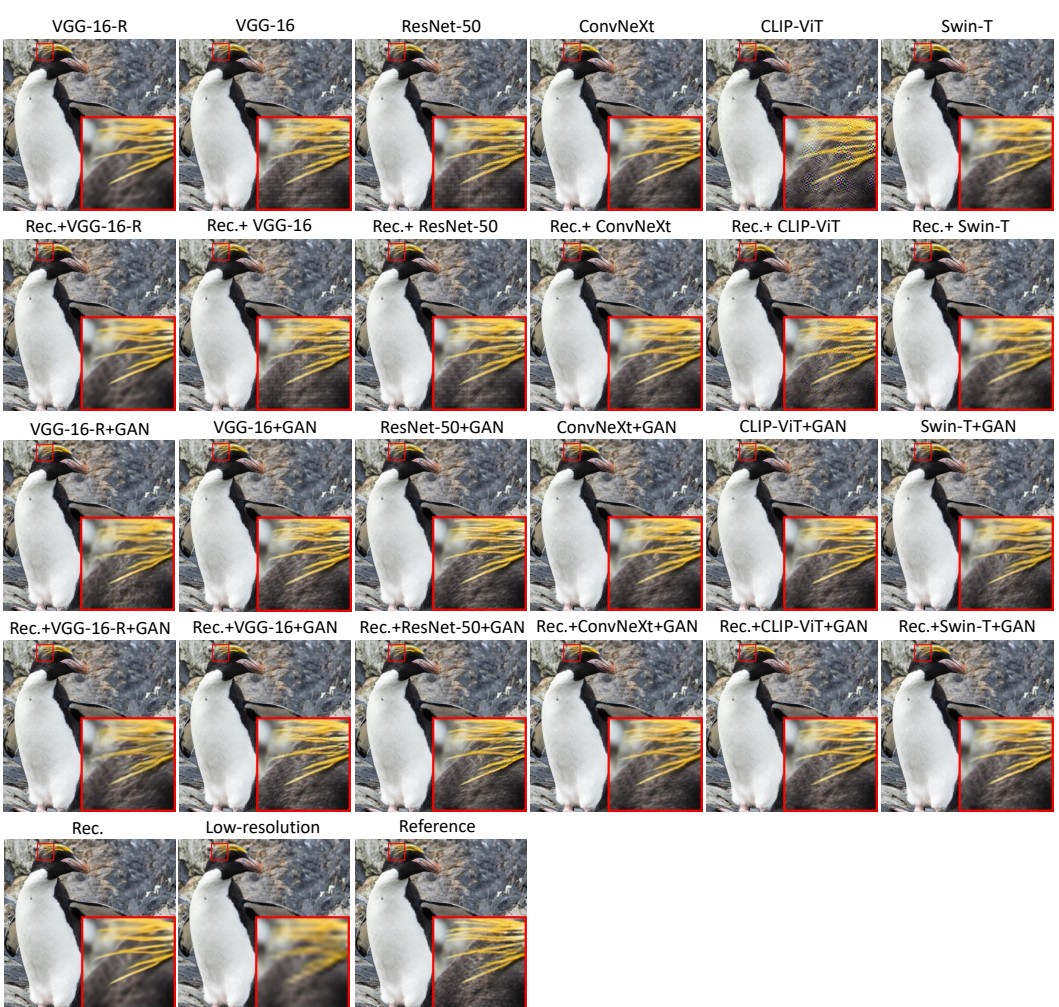

Figure 9: **Qualitative comparison** of SR results under different optimization objectives.

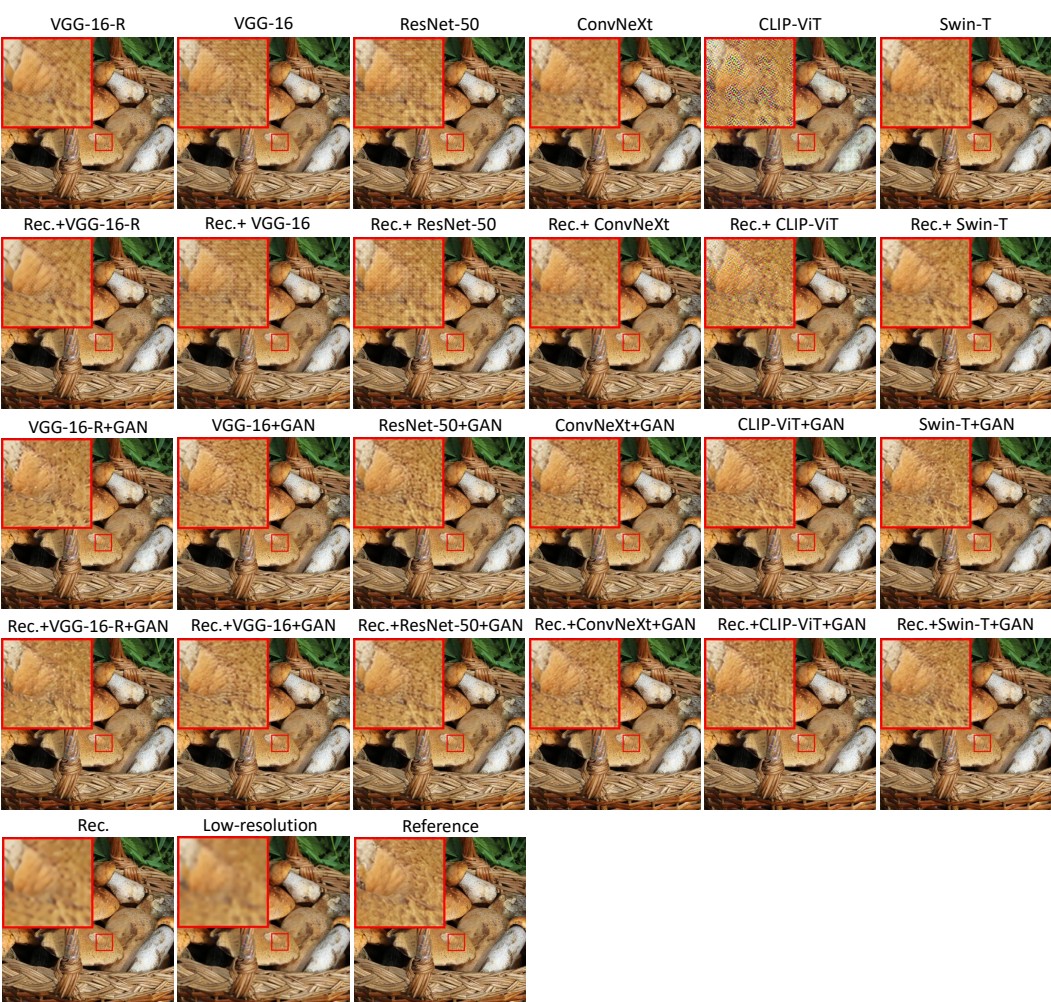

Figure 10: **Qualitative comparison** of SR results under different optimization objectives.

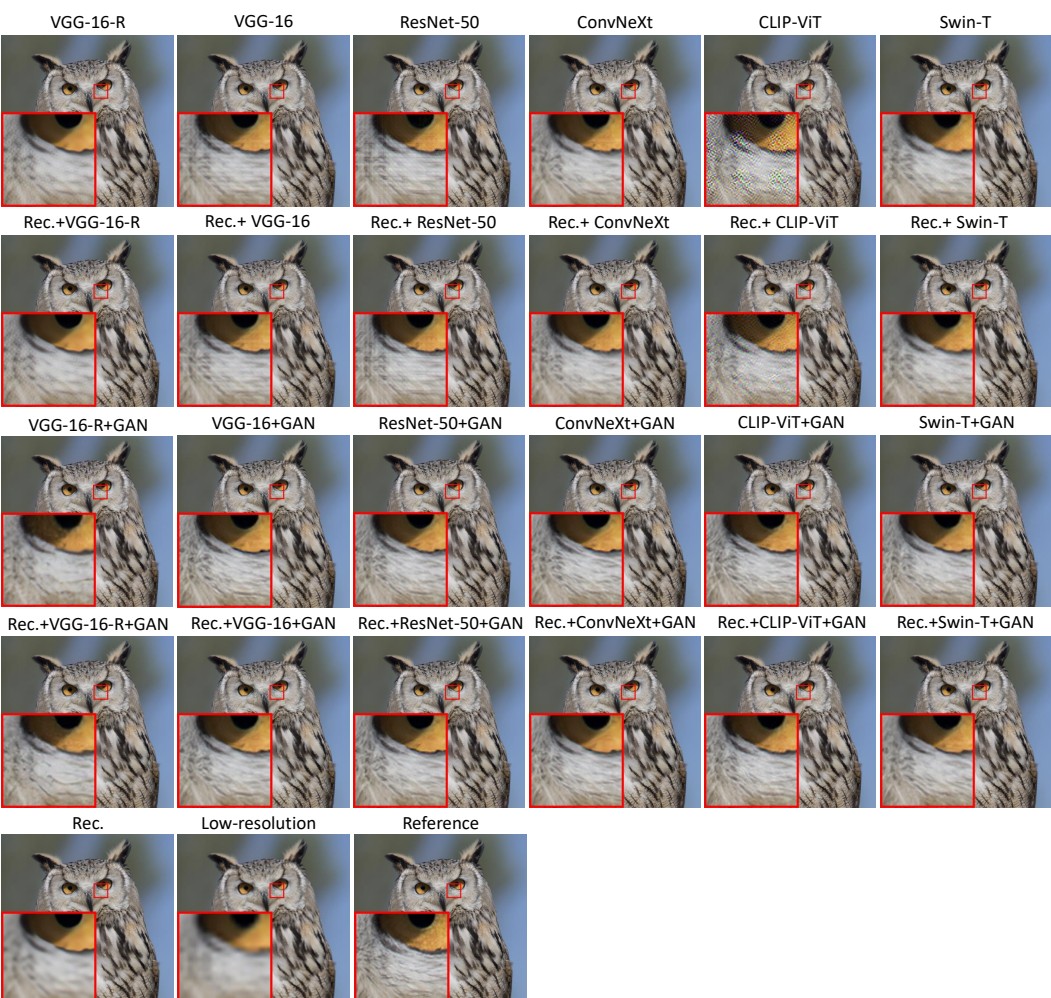

Figure 11: **Qualitative comparison** of SR results under different optimization objectives.

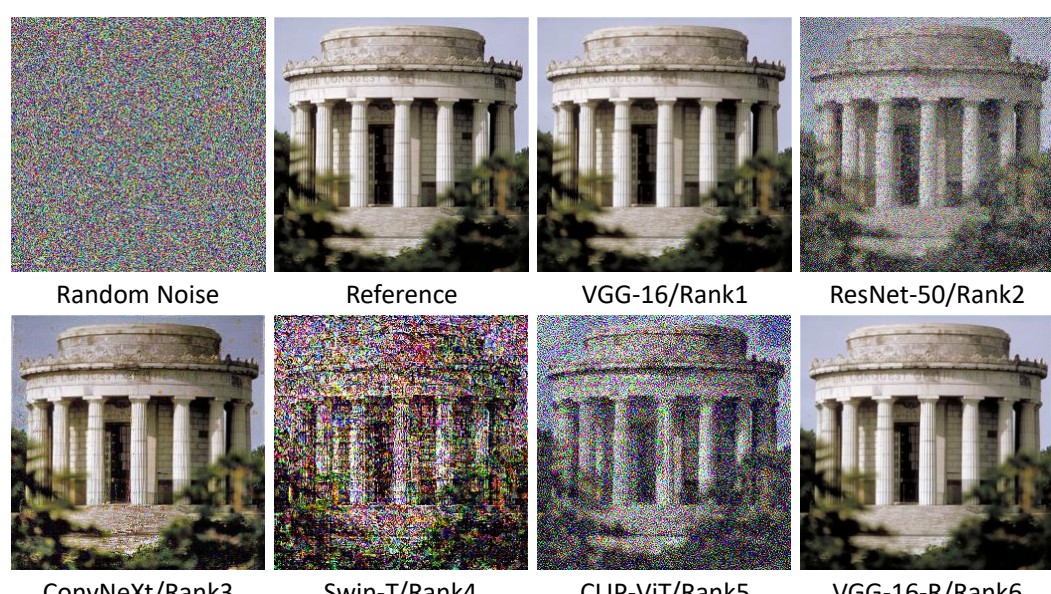

Figure 12: **Image recovery comparison** using bijection-compliant DISTS-style FR-IQA metrics

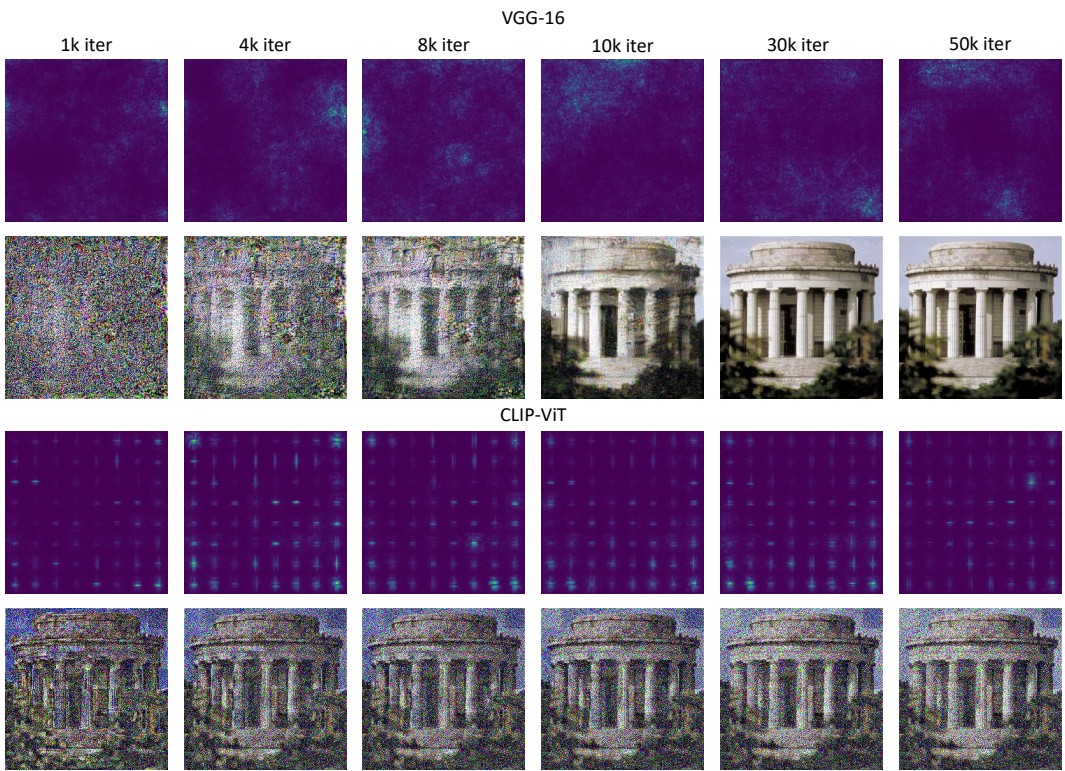

Figure 13: **Gradient visualization** via image-recovery experiments comparing VGG-16 and CLIP-ViT backbone metrics

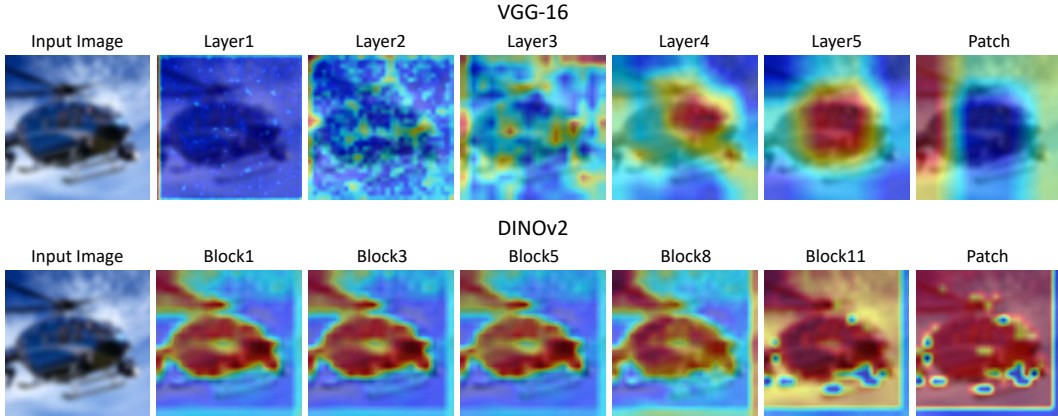

Figure 14: **Visualization of discriminator feature maps** produced by VGG-16 (top row) and DI-NOv2 (bottom row) backbones

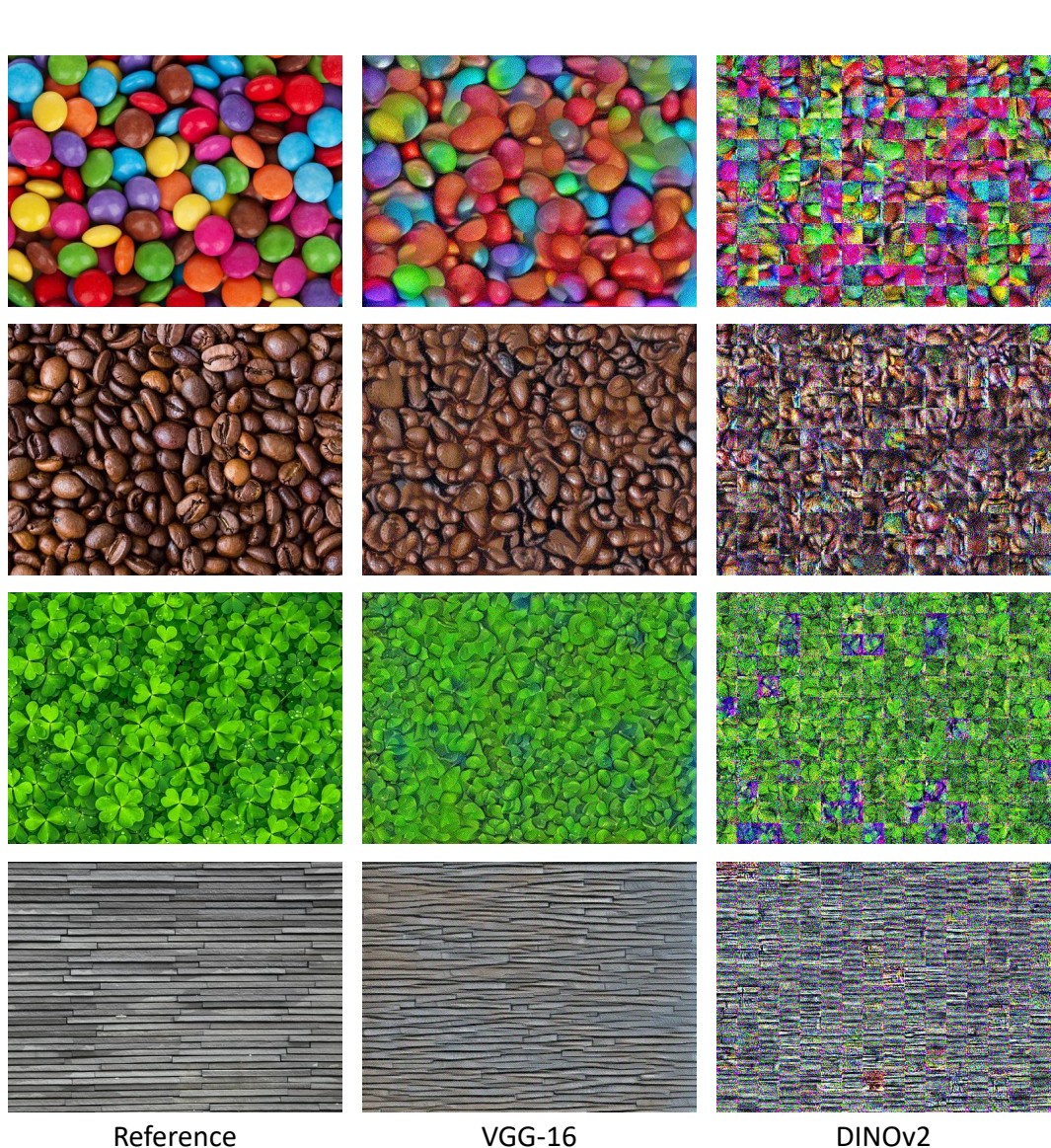

Figure 15: **Texture synthesis comparison** based on FR-IQA metrics derived from VGG-16 and DINOv2 backbones

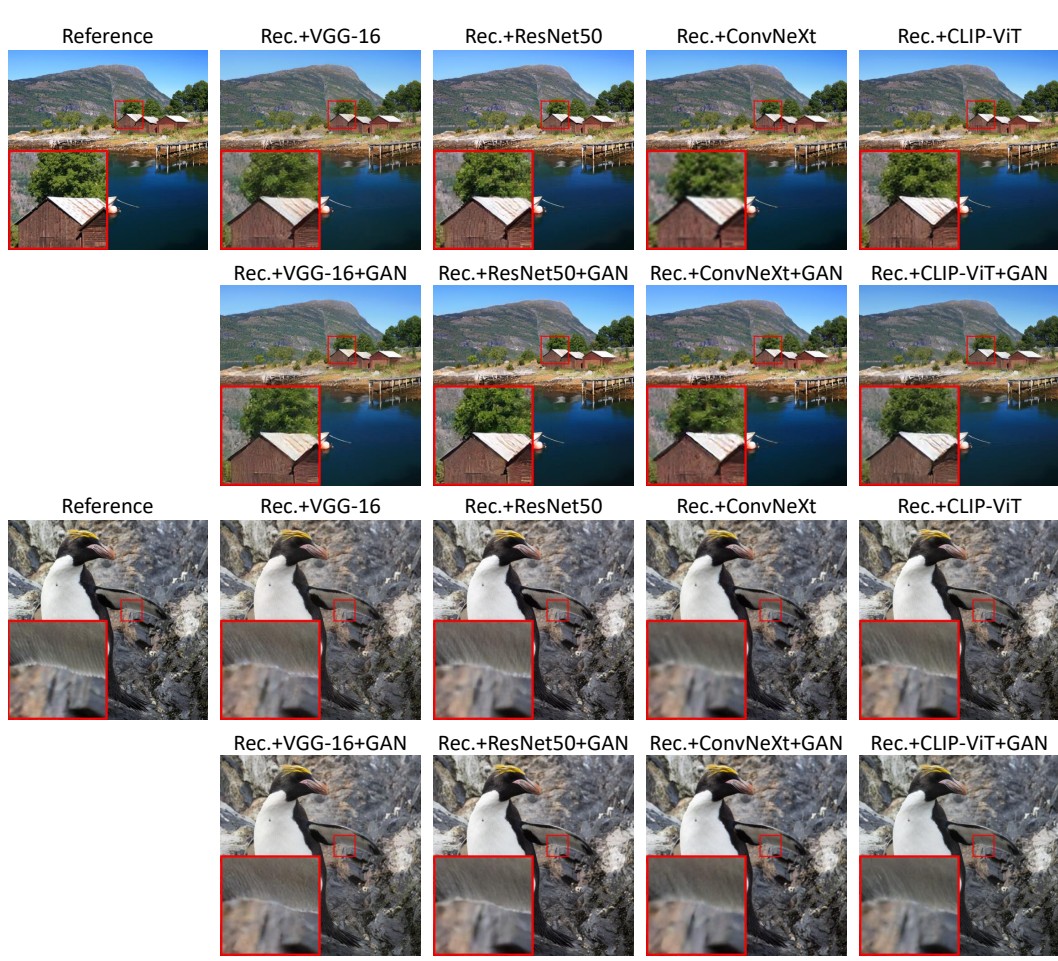

Figure 16: **Qualitative comparison** of AutoEncoder results across varying optimization objectives.

