# OpenReview forum: "The Unanticipated Asymmetry Between Perceptual Optimization and Assessment"
_ICLR.cc/2026/Conference — Submitted to ICLR 2026_

### Official Review · Reviewer_EoiF · 2025-10-24

**Soundness:** 2
**Presentation:** 2
**Contribution:** 2
**Rating:** 4
**Confidence:** 3

**Summary:**

This paper investigates the relationship and potential asymmetry between perceptual optimization objectives and their effectiveness as image quality assessment (IQA) metrics in image super-resolution (SR) tasks. The authors construct a family of DISTS-style perceptual metrics with diverse backbone architectures, design an array of adversarial discriminators, and perform comprehensive experiments on SwinIR. They report that the best-performing IQA metrics do not necessarily yield superior perceptual optimization, and adversarially trained discriminators, while effective at artifact suppression, do not transfer as strong initializations for IQA models. The work highlights the critical influence of discriminator design, particularly patch-level convolutional architectures, on optimization results and training stability.

**Strengths:**

1. Comprehensive Experimental Evaluation: The paper presents an controlled empirical study, spanning multiple backbone architectures (VGG-16, ResNet-50, ConvNeXt, CLIP-ViT, Swin-T, and DINOv2) for both perceptual losses and discriminators. This breadth adds significant credibility to the claims.
2. Systematic Analysis of Loss Design: The explicit formulation and analysis of the DISTS-style perceptual metric and its alternative feature backbones is thorough, with concrete mathematical details provided.
3. Insight into Optimization-Assessment Asymmetry: The study surfaces a well-supported, thought-provoking observation that high IQA performance does not guarantee optimization efficacy.
4. Strong Visualizations: The qualitative figures are well-chosen and annotated, highlighting subtle texture and artifact differences that strengthen the analysis.

**Weaknesses:**

1. Limited Generalization Beyond SR and SwinIR: The study is conducted entirely within the SwinIR SR framework, with all optimization experiments targeting one architecture. There is no evidence showing similar asymmetries arise for other image-to-image or generative tasks.
2. Superficial Discussion of Optimization Failures: While the work documents the underwhelming performance of some metrics in optimization, it rarely delves into why this happens.
3. Insufficient Theoretical Context: While the empirical results are robust, the theoretical motivation for the observed asymmetry is shallow.
4. Overreliance on NR-IQA for Main Evaluation: Table 1 uses NR-IQA scores as the primary measure for comparing perceptual optimization effectiveness. While defensible given the issues noted for FR-IQA, this risks circularity: the choice of NR metric may bias the conclusions, especially if the NR metric is itself optimized using similar backbones.

**Questions:**

1. On Generalization: Can the observed asymmetry between optimization and assessment be demonstrated for tasks beyond SR or architectures beyond SwinIR?
2. Metric Structure and Failure Analysis: Have you conducted feature visualizations or layer-wise analyses to pinpoint why certain backbones or perceptual metrics perform poorly during optimization?
3. NR-IQA Metric Bias: Your evaluation heavily relies on a set of modern NR-IQA methods. How sensitive are your findings to this choice? Did you observe substantially different trends using alternative NR or FR-IQA metrics?
4. Explaining Training Instabilities (Figure 7): Do you have further diagnostics explaining why DINOv2-based discriminators degrade at higher GAN weights, while ResNet-50 remains robust?

---

> ### Author Response · Authors · 2025-11-26
>
> We sincerely thank this reviewer for the insightful feedback and the thorough evaluation of our manuscript. Our newly uploaded paper contains only the figures required for addressing the reviewers’ questions. We have highlighted the figure captions in blue, and all experimental analyses are included in the rebuttal to make it easier for the reviewers to examine our responses. We hope our responses can address this reviewer's concerns.
>
>
> **`[W1 & Q1] Regarding Generalization`**
>
> Thanks for the valuable and constructive feedback. To ensure that the observed asymmetry is not specific to SwinIR or the SR task itself, we expanded our experiments under both non-adversarial and adversarial optimization in two directions. First, we evaluated additional architectures, including the CNN-based SRResNet and the Transformer–CNN hybrid HAT; for detailed experimental settings and analysis, kindly refer to our response to **`Reviewer 22QZ [W1]`**. Second, we extended the investigation across tasks by introducing an auto-encoder reconstruction task, which serves as a fundamental compression task and provides a basic cornerstone for generative applications while using the same six constructed bijective FR-IQA metrics. The autoencoder architecture matches that of the SD3 autoencoder and was trained on the DIV2K dataset. The overall results are summarized in the table below.
>
>
> \begin{array}{l|cccccc}
> \hline
> \text{Metric} & \text{VGG-16-R} & \text{VGG-16} & \text{ResNet-50} & \text{ConvNeXt} & \text{CLIP-ViT} & \text{Swin-T} \newline
> \hline
> \textit{Reconstruction and Perceptual} & & & & & & \newline
> \text{MANIQA} & 28.13 & 30.48 & 34.54 & 25.62 & 42.01 & 21.97 \newline
> \text{LIQE} & 53.97 & 57.18 & 59.68 & 39.34 & 60.66 & 35.87 \newline
> \text{DeQA{-}Score} & 54.24 & 52.43 & 54.89 & 36.71 & 50.71 & 32.95 \newline
> \text{VisualQuality{-}R1} & 39.73 & 55.53 & 55.99 & 33.96 & 51.44 & 27.83 \newline
> \text{Average} & 44.02 & 48.91 & 51.28 & 33.91 & 51.21 & 29.66 \newline \hline
> \text{IQA Rank} & 6 & 1 & 2 & 3 & 5 & 4 \newline
> \hline
> \textit{Reconstruction, Perceptual and Adversarial} & & & & & & \newline
> \text{MANIQA} & 29.91 & 33.46 & 35.47 & 29.17 & 41.12 & 23.22 \newline
> \text{LIQE} & 51.69 & 60.40 & 62.28 & 48.11 & 61.22 & 35.62 \newline
> \text{DeQA{-}Score} & 46.91 & 54.94 & 55.58 & 42.04 & 53.17 & 30.53 \newline
> \text{VisualQuality{-}R1} & 38.78 & 52.92 & 53.76 & 39.22 & 51.23 & 25.12 \newline
> \text{Average} & 41.82 & 50.43 & 51.77 & 39.64 & 51.69 & 28.62 \newline \hline
> \text{IQA Rank} & 6 & 1 & 2 & 3 & 5 & 4 \newline
> \hline
> \end{array}
>
>
>
> **Result Analysis.** Our autoencoder experiments reveal that the same asymmetry observed in the SR task persists: high standalone IQA performance does not reliably correlate with optimization effectiveness during training. However, compared with the SR task, the effect of adversarial loss behaves differently. In SR, adding adversarial loss often yields substantial improvements under multiple loss combinations (as shown in Table 1 of the main paper). In contrast, for the autoencoder reconstruction task, adversarial loss provides only modest gains (e.g., for VGG-16) and fails to consistently improve performance across all perceptual-loss configurations. This suggests that adversarial optimization interacts more weakly with the autoencoder objective, likely due to the simpler, more deterministic reconstruction nature of autoencoders.
>
> Furthermore, by comparing the results in the table with those in Table 1 of the main paper, we find that the choice of perceptual metric has a much greater impact on reconstruction quality in the autoencoder setup. For instance, metrics derived from Swin-T and ConvNeXt lead to reconstruction behaviors notably different from those based on VGG or ResNet features. This indicates that, in autoencoder training, model optimization is more sensitive to the architectural properties of the perceptual metric.
>
> These new findings reinforce our core insight: **strong IQA performance alone is insufficient for guiding perceptual optimization**. Effective perceptual training requires careful attention not only to the metric’s evaluation performance but also to metric architecture design (kindly see our response to **`Reviewer 22QZ [W1]`**) and its task adaptability. We sincerely thank the reviewer for encouraging us to explore additional tasks; all results and analyses from the autoencoder experiments will be included in the revised version of the paper.

---

> ### Author Response · Authors · 2025-11-26
>
> **`[W2, W3 & Q2] Regarding the Failure Analysis and Theoretical Context`**
>
> We thank the reviewer for their feedback regarding the failure analysis and theoretical context. We kindly refer to our response to **`Reviewer 22QZ [W2-1]`**, where we provide an in-depth analysis of why high-performing FR-IQA metrics often fail during optimization, supported by new experiments and gradient-flow visualizations. In summary, our findings reveal that optimization failures are fundamentally rooted in the **architecture-dependent gradient geometry** of deep convolutional and transformer-based models. Specifically, through controlled **image recovery experiments**, we show that:
> - **IQA accuracy does not correlate with optimization behavior**---metrics with the highest FR-IQA performance (e.g., ConvNeXt, CLIP-ViT, Swin-T) frequently fail to reconstruct even a coarse structure, while the lowest-performing metric (VGG-16-R) succeeds.
> - Although all metrics are theoretically **bijective**, deeper architectures introduce complex multi-layer alignment constraints that distort the mapping between pixel perturbations and feature-space discrepancies, leading to **poor local minima and vanishing / irregular gradients**, thus preventing convergence in practice.
> - Our **gradient-field** visualizations further reveal the structural cause: VGG-based metrics produce smooth, spatially coherent gradients that form a stable optimization landscape, whereas transformer-based metrics generate block-like, fragmented gradients induced by patch embeddings—producing highly irregular descent directions that destabilize optimization.
>
> These analyses provide the requested theoretical context, architecture-level explanation, and visualization-based evidence for why certain backbones exhibit asymmetric behavior between “IQA evaluation” and “perceptual optimization.” We will incorporate this expanded theoretical motivation and failure analysis into the revised manuscript.

---

> ### Author Response · Authors · 2025-11-26
>
> **`[W4 & Q3] Regarding the Use of NR-IQA for Main Evaluation`**
>
> Thanks for the thoughtful comments. We address concerns below.
>
> **Regarding the potential circularity** of relying on NR-IQA metrics, we emphasize that our evaluation setup is independent and does not bias the conclusions. (1) Using NR-IQA models as perceptual evaluation metrics is a widely adopted practice in both high-level [7] and low-level vision tasks [8], and continues to drive progress in assessing visual quality. (2) Although some NR-IQA models may share architectural components with modern SR networks, their training data, optimization objectives, and task formulations are entirely different. Therefore, the evaluation does not favor our method in any systematic way. (3) To further verify the reliability of NR-IQA-based evaluation, we conducted an additional user study. Please kindly refer to **`Reviewer iVQ1 [W2]`**. The results consistently show that the advanced NR-IQA metrics we choose exhibit strong alignment with human perceptual judgments.
>
>
> **Regarding the sensitivity to the choice of NR-IQA metrics**: in this work we carefully select four advanced and widely used NR-IQA models. VisualQuality-R1 and DeQA-Score leverage MLLM backbones, while LIQE and MANIQA are classical and broadly adopted NR-IQA methods, widely used in applications such as video-generation reward/benchmarking [9], data filtering [10], and general low-level evaluation [8]. Our findings remain robust across these diverse NR-IQA families, indicating that the conclusions are not driven by metric-specific bias.
>
>
> **Regarding the use of different NR-IQA metrics.** We argue that it is preferable to adopt accurate NR-IQA metrics when evaluating super-resolved images. As shown in `Fig. 4`, different perceptual metrics indeed tend to guide SR models toward generating visually distinguishable results. Even relatively early metrics such as NIQE and MUSIQ are still capable of providing reasonably accurate judgments. To further clarify this point, we additionally evaluated NIQE and MUSIQ through a user study (details are provided in our response to **`Reviewer iVQ1 [W2]`**). The corresponding results are presented in the table below.
>
>
> \begin{array}{l|ccccccc}
> \hline
> \text{Metric} & \text{Avg{-}rating} & \text{VQ{-}R1} & \text{DeQA} & \text{LIQE} & \text{MANIQA} & \text{MUSIQ} & \text{NIQE} \newline
> \hline
> \text{Accuracy} &0.93  & 0.88 & 0.82 & 0.84 & 0.90 &0.81 &0.71  \newline
> \hline
> \end{array}
>
>
> As shown in the table above, both NIQE and MUSIQ also provide reasonably accurate quality estimations, achieving accuracies of 0.71 and 0.81, respectively. Importantly, they exhibit the same qualitative trend as the four selected NR-IQA metrics: when comparing super-resolved images generated by a non-adversarial model versus those produced with adversarial loss, their judgments remain consistent with the other perceptual metrics. This further supports our argument that, although newer NR-IQA models tend to yield higher absolute accuracy, earlier metrics such as NIQE and MUSIQ are still capable of reliably reflecting the perceptual differences between the two types of SR outputs.
>
>
>
>
> [7] Liu, J., Liu, G., Liang, J., Li, Y., Liu, J., Wang, X., ... & Ouyang, W. (2025). Flow-GRPO: Training flow matching models via online RL. arXiv preprint arXiv:2505.05470.
>
> [8] Lin, X., He, J., Chen, Z., Lyu, Z., Dai, B., Yu, F., ... & Dong, C. (2024, September). Diffbir: Toward blind image restoration with generative diffusion prior. In European Conference on Computer Vision (pp. 430-448).
>
> [9] Wan, T., Wang, A., Ai, B., Wen, B., Mao, C., Xie, C. W., ... & Liu, Z. (2025). Wan: Open and advanced large-scale video generative models. arXiv preprint arXiv:2503.20314.
>
> [10] Yang, Y., Tian, H., Shi, Y., Xie, W., Zhang, Y. F., Dong, Y., ... & Tan, T. (2025). A Survey of Unified Multimodal Understanding and Generation: Advances and Challenges. Authorea Preprints.

---

> ### Author Response · Authors · 2025-11-26
>
> **`[Q4] Regarding The Explanation of Training Instabilities`**
>
> We sincerely appreciate the reviewer’s thoughtful comments. As previously discussed (kindly refer to **`Reviewer 22QZ [W2-2]`**), our additional diagnostics consistently show that the degradation of DINOv2-based discriminators at higher GAN weights arises from **intrinsic architectural differences rather than training configurations**.
> - Transformer (DINOv2) discriminators exhibit globally diffused and weakly localized activations, making their PatchGAN outputs lose spatial discriminativeness and reducing their sensitivity to local artifacts while amplifying global noise.
> - Because global self-attention lacks inherent locality priors, Transformer discriminators produce unstable updates when attention shifts across distant patches, which becomes increasingly problematic under stronger GAN losses.
> - In contrast, CNN-based discriminators such as VGG and ResNet-50 generate localized, high-frequency, and structurally interpretable activations that provide stable, spatially grounded gradients throughout training.
>
> Altogether, these analyses show that the instability of Transformer-based discriminators arises from fundamental architectural limitations rather than hyperparameter settings, whereas CNN architectures naturally offer the locality and inductive biases required for stable and effective adversarial learning. We will include the results in the revised manuscript.

---

### Official Review · Reviewer_iVQ1 · 2025-10-28

**Soundness:** 4
**Presentation:** 4
**Contribution:** 4
**Rating:** 8
**Confidence:** 5

**Summary:**

This paper reveals a surprising mismatch between image quality assessment (IQA) metrics and their usefulness as training objectives: metrics that score highly for IQA don’t necessarily improve perceptual optimization, and this gap widens with adversarial training. It also finds that while GAN discriminators help suppress artifacts and boost realism, especially patch-level, convolutional designs, their learned features transfer poorly when reused to initialize IQA models. Overall, adversarial supervision tends to dominate outcomes once a reasonable perceptual loss is present, challenging the common practice of equating IQA strength with optimization effectiveness.

**Strengths:**

1.	This finding shows that high-scoring IQA metrics do not necessarily make better optimization objectives, overturning a long-held assumption and opening a new direction for perceptual loss design.
2.	The study presents a carefully controlled experimental framework spanning multiple loss compositions, discriminator architectures, and perceptual backbones, ensuring robust and reproducible findings.
3.	The paper is clearly written, with concise exposition and high readability.

**Weaknesses:**

1.	The analysis should be supplemented with results on additional architectures such as HAT and other models to verify whether the same properties persist beyond the SwinIR-based cases.
2.	The paper should include a user study to mitigate potential inaccuracies of NR-IQA scores and verify that the observed improvements align with human judgments.
3.	Figure 7 should include additional GAN loss weights (at least six) to better illustrate the effects.

**Questions:**

N.A.

---

> ### Author Response · Authors · 2025-11-26
>
> We sincerely thank this reviewer for the insightful feedback and the thorough evaluation of our manuscript. Our newly uploaded paper contains only the figures required for addressing the reviewers’ questions. We have highlighted the figure captions in blue, and all experimental analyses are included in the rebuttal to make it easier for the reviewers to examine our responses. We hope our responses can address this reviewer's concerns.
>
>
> **`[W1] Regarding Additional Architectures`**
>
> Thanks for the constructive comment. We have also evaluated the representative CNN based model SRResNet and the Transformer CNN hybrid HAT to further validate the asymmetry. For the detailed experimental settings and analysis of the results, please kindly refer to our response to **`Reviewer 22QZ [W1]`**. These results will be included in the revised version of the paper.
>
>
> **`[W2] Regarding User Study`**
>
> Thanks for the valuable and constructive feedback. We conducted user studies in which, for each FR-IQA metric, 30 image pairs were randomly selected to compare the configuration using **combined reconstruction and perceptual losses** with the configuration using **reconstruction, perceptual, and adversarial losses**, resulting in a total of 180 pairs across six FR-IQA metrics. Each pair was evaluated by five participants, and the final label was determined through majority voting. Participants viewed each pair side by side with unrestricted zooming and panning and selected the image with higher visual quality. The resulting accuracy scores are presented in the table below, where Avg-rating refers to rating image quality using the average score across VisualQuality-R1, DeQA-Score, LIQE, and MANIQA.
>
> \begin{array}{l|ccccccc}
> \hline
> \text{Metric} & \text{Avg{-}rating} & \text{VQ{-}R1} & \text{DeQA} & \text{LIQE} & \text{MANIQA}  \newline
> \hline
> \text{Accuracy} &0.93  & 0.88 & 0.82 & 0.84 & 0.90  \newline
> \hline
> \end{array}
>
> Based on the results shown in the table above, all selected NR-IQA metrics achieve accuracy scores above 80\%. Moreover, the Avg-rating metric, obtained by averaging VisualQuality-R1, DeQA-Score, LIQE, and MANIQA, reaches an accuracy of 93\%. These findings demonstrate that the chosen advanced NR-IQA metrics exhibit strong consistency with human perceptual judgments. We will include the results in the revised manuscript.
>
>
> **`[W3] Regarding Additional GAN Loss Weights`**
>
> Thanks for raising this important point. We have extended the adversarial weight range from the original setting to $[5\times10^{-4}, 1\times10^{-1}]$, covering a total of six additional weight values. The updated results are shown in the table below.
>
> \begin{array}{l|cccccc|c}
> \hline
> \text{Metric} & \text{5e-4} & \text{1e-3} & \text{5e-3} & \text{1e-2} & \text{5e-2} & \text{1e-1} &\text{Std.} \newline
> \hline
> \textit{RN50 PatchGAN} & & & & & & \newline
> \text{MANIQA} & 58.70 & 58.82 & 58.87 & 58.93 & 58.31 & 58.36 & 0.243\newline
> \text{LIQE} & 68.06 & 68.32 & 68.24 & 68.36 & 67.92 & 68.86 & 0.294\newline
> \text{DeQA-Score} & 66.90 & 66.51 & 66.05 & 65.50 & 64.31 & 64.49 & 0.969\newline
> \text{VisualQuality-R1} & 62.69 & 62.44 & 62.51 & 62.21 & 62.00 & 61.72 & 0.326\newline
> \hline
> \text{Average} & 64.09 & 64.02 & 63.92 & 63.75 & 63.14 & 63.36 &0.350\newline
> \hline
> \textit{DINOv2 PatchGAN} & & & & & & \newline
> \text{MANIQA} & 54.90 & 55.07 & 57.79 & 57.32 & 35.07 & 29.91 & 11.36\newline
> \text{LIQE} & 59.67 & 59.81 & 67.57 & 67.38 & 57.15 & 54.35 & 4.934\newline
> \text{DeQA-Score} & 63.25 & 62.78 & 62.74 & 62.78 & 49.98 & 46.66 & 6.937\newline
> \text{VisualQuality-R1} & 60.53 & 60.38 & 61.22 & 61.18 & 51.33 & 46.54 & 5.782\newline
> \hline
> \text{Average} & 59.59 & 59.51 & 62.33 & 62.16 & 48.38 & 44.37 & 7.030\newline
> \hline
> \textit{VGG-16 PatchGAN} & & & & & & \newline
> \text{MANIQA} & 59.35 & 59.05 & 57.74 & 55.97 & 51.40 & 44.69 & 5.203\newline
> \text{LIQE} & 68.21 & 67.94 & 67.72 & 67.45 & 64.97 & 63.67 & 1.711\newline
> \text{DeQA-Score} & 63.40 & 63.36 & 63.20 & 61.14 & 58.49 & 56.78 & 2.592\newline
> \text{VisualQuality-R1} & 62.02 & 61.91 & 61.79 & 60.59 & 56.57 & 53.93 & 3.115\newline
> \hline
> \text{Average} & 63.25 & 63.06 & 62.61 & 61.29 & 57.86 & 54.77 & 3.137\newline
> \hline
> \end{array}
>
>
> **Result Analysis.** The table clearly shows that convolution-based discriminators, particularly the ResNet-50 discriminator, consistently achieve higher SR scores across a wide range of weights and exhibit lower variance. This suggests that compared with ViT-based discriminators, convolutional discriminators are inherently more stable, likely due to their stronger inductive biases and locality modeling. These observations also align with our response to **`Reviewer 22QZ [W2-2]`**. We will include the results in the revised manuscript.

---

> > ### Comment · Reviewer_iVQ1 · 2025-11-27
> >
> > Having carefully read the authors’ response to my review and to the comments from the other two reviewers, especially regarding the theoretical analysis, I would like to thank the authors for their thorough rebuttal. I sincerely recommend that the theoretical analysis be revised and incorporated into the main paper, as it is an essential core of this work. My concerns have been fully addressed, and I am willing to maintain my original score.

---

### Official Review · Reviewer_22QZ · 2025-10-31

**Soundness:** 2
**Presentation:** 2
**Contribution:** 2
**Rating:** 4
**Confidence:** 2

**Summary:**

This paper conducts systematic experiments to investigate the relationship between perceptual optimization and image quality assessment. Using image-super-resolution with SwinIR as the testbed, this work constructs and evaluates a family of DISTS-style perceptual metrics using various backbones and examines whether discriminator features learned during adversarial training can transfer to IQA tasks. Meanwhile, analyses on the choices of GAN-based initialization and architecture of discriminator are provided.

**Strengths:**

The study systematically evaluates multiple backbones, discriminator designs, and loss combinations, revealing that good IQA metrics are not necessarily effective for perceptual optimization.

**Weaknesses:**

1. The experiments are restricted to super-resolution using SwinIR. As this study focuses on revealing the effectiveness of different IQA metrics and discriminator designs, the choice of backbone architecture is necessary and should be considered.
2. Although the asymmetry is clearly demonstrated empirically, there is limited theoretical analysis explaining why high-performing IQA metrics fail in optimization. Meanwhile, the results in Sec 3.4 suggest that Transformer-based discriminators perform poorly and are more unstable. However, it lacks detailed analysis on attention behaviors or feature maps.
3. This work has limited practical guidance. For instance, if IQA metrics and optimization are misaligned, what practical heuristics should be followed when selecting losses?
4. In Table 2, the extremely low performance of DINOv2 using ImageNet initialization on TQD is not explained.

**Questions:**

Please refer to the weakness part.

---

> ### Author Response · Authors · 2025-11-26
>
> We sincerely thank this reviewer for the insightful feedback and the thorough evaluation of our manuscript. Our newly uploaded paper contains only the figures required for addressing the reviewers’ questions. We have highlighted the figure captions in blue, and all experimental analyses are included in the rebuttal to make it easier for the reviewers to examine our responses. We hope our responses can address this reviewer's concerns.
>
> **`[W1] Regarding Backbone Architecture`**
>
> Thanks for the constructive comment. We also conducted experiments using two additional widely adopted super-resolution (SR) methods: the CNN-based SRResNet and the Transformer-CNN hybrid HAT. Our evaluation focuses on the two representative optimization settings described in the main paper (lines 246–247): (1) a combination of reconstruction and perceptual losses, and (2) the full objective in Eq. 1, which includes reconstruction, perceptual, and adversarial losses. The results are summarized in the table below.
>
> \begin{array}{l|cccccc|c|c|c}
> \hline
> \text{Metric} & \text{VGG-16-R} & \text{VGG-16} & \text{ResNet-50} & \text{ConvNeXt} & \text{CLIP-ViT} & \text{Swin-T} & \text{Std.} & \ell_1 & \text{Ref.} \newline
> \hline
> \textit{HAT. Reconstruction and Perceptual} & & & & & & & & & \newline
> \text{MANIQA} & 50.33 & 56.31 & 51.48 & 56.93 & 54.67 & 51.72 & 2.526 & 48.50 & 61.78 \newline
> \text{LIQE} & 57.79 & 60.99 & 59.08 & 64.64 & 65.89 & 61.27 & 2.858 & 54.57 & 67.50 \newline
> \text{DeQA} & 62.69 & 64.07 & 64.48 & 63.51 & 59.14 & 55.26 & 3.303 & 53.56 & 64.96 \newline
> \text{VQ{-}R1} & 59.59 & 61.55 & 60.88 & 61.52 & 59.19 & 56.66 & 1.703 & 51.31 & 63.21 \newline
> \text{Average} & 57.60 & 60.73 & 58.98 & 61.65 & 59.72 & 56.23 & 1.827 & 51.98 & 64.36 \newline\hline
> \text{IQA Rank} & 6 & 1 & 2 & 3 & 5 & 4 & - & - & - \newline
> \hline
> \textit{HAT. Reconstruction, Perceptual and Adversarial} & & & & & & & & & \newline
> \text{MANIQA} & 59.83 & 59.17 & 59.97 & 59.84 & 59.56 & 59.51 & 0.267 & 48.50 & 61.78 \newline
> \text{LIQE} & 68.01 & 68.17 & 68.64 & 67.65 & 67.74 & 67.80 & 0.334 & 54.57 & 67.50 \newline
> \text{DeQA} & 64.48 & 63.16 & 63.72 & 63.49 & 62.72 & 62.97 & 0.574 & 53.56 & 64.96 \newline
> \text{VQ{-}R1} & 62.74 & 62.02 & 62.24 & 62.22 & 62.17 & 61.62 & 0.331 & 51.31 & 63.21 \newline
> \text{Average} & 63.77 & 63.13 & 63.64 & 63.30 & 63.05 & 62.98 & 0.297 & 51.98 & 64.36 \newline\hline
> \text{IQA Rank} & 6 & 1 & 2 & 3 & 5 & 4 & - & - & - \newline
> \hline
> \textit{SRResNet. Reconstruction and Perceptual} & & & & & & & & & \newline
> \text{MANIQA} & 43.53 & 47.71 & 44.28 & 46.08 & 52.53 & 39.64 & 3.963 & 48.50 & 61.78 \newline
> \text{LIQE} & 56.33 & 55.42 & 55.60 & 54.57 & 65.04 & 49.11 & 4.688 & 54.57 & 67.50 \newline
> \text{DeQA} & 59.50 & 58.88 & 58.73 & 50.32 & 58.45 & 45.96 & 5.231 & 53.56 & 64.96 \newline
> \text{VQ{-}R1} & 56.87 & 60.35 & 58.78 & 53.40 & 59.27 & 47.07 & 4.557 & 51.31 & 63.21 \newline
> \text{Average} & 54.06 & 55.59 & 54.35 & 51.09 & 58.82 & 45.45 & 4.161 & 51.98 & 64.36 \newline\hline
> \text{IQA Rank} & 6 & 1 & 2 & 3 & 5 & 4 & - & - & - \newline
> \hline
> \textit{SRResNet. Reconstruction, Perceptual and Adversarial} & & & & & & & & & \newline
> \text{MANIQA} & 43.83 & 53.58 & 54.08 & 52.08 & 52.45 & 44.41 & 4.264 & 48.50 & 61.78 \newline
> \text{LIQE} & 64.94 & 67.56 & 68.25 & 65.40 & 67.24 & 63.03 & 1.794 & 54.57 & 67.50 \newline
> \text{DeQA} & 59.96 & 61.70 & 62.91 & 58.82 & 62.40 & 55.55 & 2.517 & 53.56 & 64.96 \newline
> \text{VQ{-}R1} & 57.82 & 60.31 & 60.94 & 58.69 & 60.84 & 55.66 & 1.893 & 51.31 & 63.21 \newline
> \text{Average} & 56.64 & 60.79 & 61.55 & 58.75 & 60.72 & 54.66 & 2.483 & 51.98 & 64.36 \newline\hline
> \text{IQA Rank} & 6 & 1 & 2 & 3 & 5 & 4 & - & - & - \newline
> \hline
> \end{array}
>
>
> **Result Analysis.** Compared with the SwinIR results in Table 1 of the main paper, we observe that the asymmetry generalizes across convolutional, Transformer, and hybrid SR architectures, which indicates that high IQA performance does not necessarily guide the network toward better generative outcomes. Moreover, the homogenizing effect of the adversarial loss persists: different perceptual metrics produce increasingly similar visual results, reducing performance gaps across objectives. These findings further reinforce that the asymmetry between perceptual optimization and perceptual assessment is largely insensitive to the choice of **backbone** and is instead strongly determined by the selected optimization loss. These results will be included in the revised version of the paper.

---

> ### Author Response · Authors · 2025-11-26
>
> **`[W2-1] Regarding the Failure of High-Performing IQA Metrics in Optimization`**
>
> We appreciate the reviewer's constructive comments regarding the theoretical analysis of why high-performing IQA metrics fail in optimization, which indeed touches on the core issues of perceptual optimization. In response, following the settings in [1, 2], we conduct an **image recovery** experiment as a toy example to validate the optimization difficulty and the gradient flow inherent in perceptual optimization. Specifically, image recovery is performed by solving
> $x^\star = \arg\min_x l(x, y)$, where $l(x, y)$ denotes a fixed full-reference IQA metric used as the optimization objective, $x$ is initialized with random Gaussian noise, and $y$ is a clean structured target image. During the optimization, $x$ is iteratively updated to minimize its discrepancy with $y$. We regard the recovery process as complete when $x$ shows convergence. In practice, this occurs when the IQA metric is close to zero, sufficiently small, or no longer exhibits any meaningful **reduction**.
>
> We conduct image recovery experiments using the six proposed DISTS-style FR-IQA metrics, as shown in `Fig. 12` (revised Appendix). Several noteworthy observations emerge from this toy setting.
> - **IQA performance does not correlate with optimization behavior.** Although VGG16-R exhibits the lowest FR-IQA performance among the six metrics, it is one of the only metrics that successfully guides the random noise image toward a structured and recognizable reconstruction. In contrast, metrics with much higher IQA accuracy (e.g., ConvNeXt, CLIP-ViT, Swin-T) fail to optimize the image into a meaningful structure. This confirms that good FR-IQA accuracy does not imply good optimization geometry. The ability to serve as an effective perceptual loss is thus not aligned with standard IQA benchmarks.
>
> - **“Bijective” design does not guarantee successful convergence.** All six perceptual metrics are designed to be **bijective** with respect to the target image, in the sense that $l(x,y)=0 \Longleftrightarrow x=y.$ Thus, in theory, a perfect recovery should be achievable by optimization. However, in practice, only VGG-16-R and VGG-16 successfully converge, whereas all other deeper or more complex architectures fail. We observe that: 1) Metrics involving more convolutional blocks or transformer layers tend to fall into poor local minima during optimization. 2) Although the theoretical bijection holds at the representation level, deep architectures weaken the direct correspondence between pixel-space perturbations and feature-space alignment. 3) This leads to vanishing or highly irregular gradients, preventing the optimizer from reaching the global optimum. Thus, bijectivity alone cannot ensure optimizability, especially when the architecture introduces complicated alignment constraints across many layers.
>
> - **Gradient visualization reveals architectural causes of optimization difficulty.** To further understand the divergent behaviors, we visualize the pixel-wise gradient for VGG-16 and CLIP-ViT (`Fig. 13`). The VGG-based metric produces smooth and spatially coherent gradients. These gradients form a stable optimization landscape and provide consistent signals for iterative refinement. In contrast, the ViT-based metric exhibits gradients that form block-like patterns, directly reflecting the patch-embedding operation in transformer architectures. These block artifacts disrupt spatial consistency and result in fragmented, non-smooth gradient fields, making optimization unstable and preventing the noise image from evolving toward the target structure. This further confirms that architecture-induced gradient geometry plays a crucial role in perceptual optimization.
>
> In summary, these results show that the failure of high-performing IQA metrics in optimization is fundamentally driven by the **irregular and architecture-dependent gradient geometry of deep and transformer-based models**, revealing an intrinsic limitation of existing FR-IQA designs when used as perceptual objectives. We will include this theoretical analysis in the revised manuscript.
>
> [1] Ding, K., Ma, K., Wang, S., & Simoncelli, E. P. (2020). Image quality assessment: Unifying structure and texture similarity. IEEE Transactions on Pattern Analysis and Machine Intelligence, 44(5), 2567-2581.
>
> [2] Ding, K., Ma, K., Wang, S., & Simoncelli, E. P. (2021). Comparison of full-reference image quality models for optimization of image processing systems. International Journal of Computer Vision, 129(4), 1258-1281.

---

> ### Author Response · Authors · 2025-11-26
>
> **`[W2-2] Regarding the Analysis of Discriminators`**
>
> We sincerely thank this reviewer for the insightful suggestion. In response, we include feature-map-level comparisons between Transformer-based (DINOv2) and CNN-based (VGG-16) discriminators, demonstrating that the weaker performance and instability of Transformer discriminators stem from fundamental architectural differences. To illustrate this, we visualize the feature maps from both discriminator types at each architectural stage by averaging the output channels within that stage, as presented in `Fig. 14`.
>
> **Visualization Analysis.** Feature-map visualizations clearly reveal that Transformer-based discriminators behave fundamentally differently from CNN-based ones, and these differences directly explain their inferior performance and instability.
>
> - **Transformer discriminators show globally diffused, weakly localized activations.** Across early and late blocks, DINOv2 produces smoothed, globally distributed feature maps. High-response regions propagate across the entire object and even into background areas, indicating that the model responds to global semantics rather than localized evidence. The PatchGAN output becomes uniformly high in the final stage, meaning the discriminator fails to assign spatially discriminative “real/fake’’ signals. This aligns with our quantitative findings as well as the observations in [3], where Transformer discriminators tend to hallucinate or amplify global noise patterns.
>
> - **Global self-attention lacks locality priors.** Because Transformers aggregate information via global attention, they inherently: 1) weaken sensitivity to local textures that are critical for detecting artifacts, 2) over-react to irrelevant global correlations, and 3) induce unstable GAN dynamics when attention weights shift non-locally between patches. These architectural properties prevent Transformer discriminators from enforcing fine-grained realism, leading to lower scores and unstable training behavior (`Sec. 3.4`).
>
> - **CNN discriminators provide localized, frequency-aware, and stable gradients.** The VGG feature maps show the opposite pattern. From layer 1 to layer 5, activations gradually sharpen and become more spatially localized, highlighting meaningful high-frequency structures. The CNN PatchGAN heatmap preserves interpretable spatial contrast—artifact regions exhibit high “fake’’ activations while clean regions remain low. These properties directly correspond to the superior quantitative performance (`Table 3`) and stable dynamics (`Fig. 7`).
>
> In summary, these visual and behavioral differences confirm that the performance gap arises not from training settings but from the intrinsic architectural biases of Transformers versus CNNs. We will include the results and analysis in the revised manuscript.
>
> [3] Lee, K., Chang, H., Jiang, L., Zhang, H., Tu, Z., & Liu, C. (2021). ViTGAN: Training GANs with vision Transformers. In ICLR 2022.

---

> ### Author Response · Authors · 2025-11-26
>
> **`[W3] Regarding Practical Heuristics`**
>
> Thanks for the valuable and constructive feedback. Actually, our paper not only reveals the asymmetry between IQA performance and optimization, but also provides **actionable heuristics** for designing and combining optimization losses in perceptual optimization. We summarize these practical guidelines from three aspects:
> - **Adversarial loss is essential.** It consistently suppresses artifacts and restores local textures more effectively than perceptual loss alone, as demonstrated by `Table 1` and further SRResNet and HAT experiments.
> - **Patch-level convolutional discriminators serve as a reliable default.** This aligns with the well-known instability of large Transformer-based discriminators in generative models [4, 5]. CNN-based discriminators provide robust detail reconstruction and stable training, benefiting from their intrinsic architectural biases.
> - **Transformer-based perceptual metrics or deep IQA metrics are not recommended for optimization** (excluding reinforcement learning scenarios), due to their architecture-dependent gradient geometry, which leads to suboptimal or unstable optimization behavior.
>
> We appreciate the reviewer for highlighting these important insights from the paper, and we will incorporate them into the revised manuscript.
>
> **`[W4] Regarding DINOv2 Performance on TQD`**
>
> Thanks for pointing this out. The very low performance of the ImageNet-initialized DINOv2 metric on TQD (texture database) can be attributed to DINOv2’s feature representations being largely insensitive to texture. To further clarify this, we provide additional texture-synthesis experiments following the same procedure as DISTS [1] and A-DISTS [6]. Specifically, we use the **global mean** of each extracted feature map as the **texture representation** and minimize $x^{\star}=\arg\min_x \sum_{i,j}(u_{x_{j}}^{i} - u_{y_{j}}^{i})^2$, where $y$ is the target texture image and $x^{\star}$ is the synthesized texture image obtained via gradient-descent optimization from a random initialization. Here, $u_{x_j}^{i}$ and $u_{y_j}^{i}$ denote the spatial averages of channels $x^i_j$ and $y^{i}_{j}$, respectively. We evaluate both ImageNet-initialized VGG-16 and DINOv2, and the visualization results are presented in `Fig. 15`.
>
> **Result Analysis.** From the synthesized results, it is evident that VGG-16 features enable the generation of visually coherent and perceptually pleasing texture images, whereas DINOv2 features lead to noticeable block-like distortions. These artifacts are consistent with the patch-embedding mechanism used in Transformer-based architectures, as similarly illustrated in `Fig. 13`. Overall, the findings indicate that DINOv2’s feature representations are comparatively insensitive to fine-grained texture patterns, suggesting that Transformer-based models may be unsuitable as similarity metrics for texture-focused tasks. In contrast, VGG-16 preserves local texture statistics more effectively, making it a more suitable metric for perceptual texture similarity evaluation, as reflected by its SRCC and PLCC scores of 0.715 and 0.731.
>
> [4] Esser, P., Kulal, S., Blattmann, A., Entezari, R., Müller, J., Saini, H., ... & Rombach, R. (2024, July). Scaling rectified flow transformers for high-resolution image synthesis. In Forty-first International Conference on Machine Learning.
>
> [5] Tian, K., Jiang, Y., Yuan, Z., Peng, B., & Wang, L. (2024). Visual autoregressive modeling: Scalable image generation via next-scale prediction. Advances in Neural Information Processing Systems, 37, 84839-84865.
>
> [6] Ding, K., Liu, Y., Zou, X., Wang, S., & Ma, K. (2021, October). Locally adaptive structure and texture similarity for image quality assessment. In Proceedings of the 29th ACM International Conference on Multimedia (pp. 2483-2491).

---

### Author Response · Authors · 2025-11-30

Dear program chairs, senior area chairs, area chairs, and reviewers,

We sincerely thank all reviewers and ACs for their efforts, especially under this year’s unusual circumstances. Below is a concise summary of our rebuttal and the current review status.

---

**`Summary of Rebuttal–Discussion`**

Our paper uncovers a fundamental **asymmetry** between IQA metric accuracy and their effectiveness as perceptual optimization objectives in image super-resolution.

During the rebuttal, two reviewers (22QZ, EoiF) raised substantial theoretical or conceptual concerns. Our rebuttal directly and comprehensively addresses each point. In particular, we provided an expanded theoretical analysis supported by:

- new image-recovery experiments revealing optimization difficulties,
- architectural-level explanations of irregular gradient behavior,
- gradient-field visualizations comparing CNN vs. Transformer backbones, and
- analysis of bijectivity versus convergence failure.

The rebuttal introduces extensive new evidence—including experiments on SRResNet, HAT, and an autoencoder reconstruction task; strengthened theoretical clarification; architectural diagnostics; and a user study—all of which directly address the substantive concerns raised. One reviewer fully endorses the paper after the rebuttal, and the remaining concerns have been systematically resolved.

We hope this concise summary assists the AC in re-evaluating the submission fairly under this year’s unusual review conditions. Thank you again for your time, effort, and dedication to ensuring a fair and thoughtful review process.

Best regards,

Authors

---

### Meta-Review · Area_Chair_BbyP · 2026-01-03

**Summary:**

This paper studies the relationship between perceptual optimization objectives and image quality assessment metrics in the context of image super resolution. The submission reports an empirical asymmetry: IQA metrics that perform well do not necessarily serve as good optimization objectives. The paper includes extensive experiments across different feature extractors, and discriminators, and the authors further expanded the experimental scope during rebuttal by adding backbones and analyses. The initial reviewer scores were mixed (4, 8, and 4). The AC reads the manuscript, all reviewer reports, and the authors' responses. Reviewers generally agreed that the paper is experimentally thorough and clearly written. At the same time, they raised concerns regarding the limited backbone choice in the original submission, the depth of theoretical understanding, and the broader significance of the findings. In response, the authors add experiments with additional backbones and provided more discussion and analysis.

**Reviewer Concerns:**

After reviewing the discussion of both sides, my assessment is as follows.

First, the choice of backbone network is not considered to be a fundamental weakness. From a modeling perspective, it is unlikely that switching between CNN-based architectures and modern transformer-based SR backbones such as SwinIR or HAT would lead to qualitatively different conclusions. In fact, using a simpler backbone can arguably strengthen the generality of the conclusions. But adding a CNN backbone is wtill reasonable.

However, from this same perspective, the paper adopts a loss design that is overly complex. The use of a DISTS-style feature loss already represents a highly structured and nonstandard perceptual objective, which differs substantially from classical perceptual losses, simple feature-space losses, or pixel-level objectives. In addition, the GAN loss adopted in the paper is also relatively sophisticated. Because the analysis is conducted entirely under this complex loss configuration, it becomes difficult to argue that the reported conclusions generalize to more standard or simpler perceptual optimization settings. This weakens the breadth of the claimed insights.

Second, the paper demonstrates that different feature extractors yield different optimization behaviors and concludes that strong IQA performance does not necessarily imply effectiveness for perceptual optimization. While the empirical results are sound, the core message that IQA metrics are not reliable optimization objectives has been recognized for several years. Starting from earlier work on DISTS and related perceptual metrics, it has already been established that optimizing IQA scores alone does not reliably lead to improved perceptual quality. As such, the main empirical observation, while validated by some experiments, is not fundamentally new.

More importantly, for SR and perceptual restoration tasks, human subjective judgment is the ultimate criterion. The paper does not sufficiently address which classes of metrics align better with human preference, nor does it provide a principled discussion of how the observed asymmetry should inform metric or loss design with respect to human perception. This omission represents a major limitation of the study.

The relationship among feature extractor choice, loss function design, and IQA metrics is highly complex and difficult to summarize in a small set of general rules. While different combinations may exhibit certain empirical patterns, the ultimate objective of perceptual optimization is to achieve better visual quality as perceived by humans. Given that both IQA models and perceptual loss functions are themselves highly complex and imperfect proxies, attempting to extract general principles purely from their interactions is inherently challenging. For this reason, the AC would encourage the authors to place greater emphasis on human subjective evaluation, which remains the most reliable criterion for assessing perceptual quality in SR and restoration tasks.

Finally, the AC agrees with the concerns raised by two reviewers that the paper primarily documents phenomena without proposing a unifying or generalizable theoretical framework. While the authors added additional analysis in the rebuttal, the work still falls short of delivering a theory or conceptual model that would provide actionable insight to guide future research beyond the specific experimental settings explored here. This limits the paper’s value as a source of lasting insight for the broader community.

In summary, although the paper is carefully executed and contains extensive experiments, its main conclusions are either already well understood or insufficiently generalized, and the lack of a strong theoretical or conceptual contribution significantly limits its impact. For these reasons, the AC recommends rejection.

**Reviewer Scores:**

The two negative reviewers may not change their scores. and one positive reviewer already replied to maintain the positive rating.

---

### Decision · Program_Chairs · 2026-01-26

Reject